# Muon Optimizes Under Spectral Norm Constraints

**Lizhang Chen** *lzchen@cs.utexas.edu*
*Department of Computer Science*
*University of Texas at Austin*

**Jonathan Li** *jli@cs.utexas.edu*
*Department of Computer Science*
*University of Texas at Austin*

**Qiang Liu** *lqiang@cs.utexas.edu*
*Department of Computer Science*
*University of Texas at Austin*

**Reviewed on OpenReview:** *https://openreview.net/forum?id=Blz4hjxLwU*

## Abstract

The pursuit of faster optimization algorithms remains an active and important research direction in deep learning. Recently, the MUON optimizer (Jordan et al., 2024) has demonstrated promising empirical performance, but its theoretical foundation remains less understood. In this paper, we bridge this gap and provide a theoretical analysis of MUON by placing it within the LION-$\mathcal{K}$ family of optimizers (Chen et al., 2024). Specifically, we show that MUON corresponds to LION-$\mathcal{K}$ when equipped with the nuclear norm, and we leverage the theoretical results of LION-$\mathcal{K}$ to establish that MUON (with decoupled weight decay) implicitly solves an optimization problem that enforces a constraint on the spectral norm of weight matrices. This perspective not only demystifies the implicit regularization effects of MUON but also leads to natural generalizations through varying the choice of convex map $\mathcal{K}$, allowing for the exploration of a broader class of implicitly regularized and constrained optimization algorithms.

## 1 Introduction

Optimization remains an important research direction in deep learning, where the backpropagation algorithm (LeCun, 1989) enables efficient and scalable gradient-based training of neural architectures. Among gradient-based optimizers, adaptive methods such as ADAGRAD (Duchi et al., 2011), ADAM (Kingma & Ba, 2015), and ADAMW (Loshchilov & Hutter, 2019) have become standard for training large-scale deep neural networks due to their ability to dynamically adjust learning rates based on first- and second-order moment estimates.

Recent advances in optimization algorithms have shown promising potential to outperform traditional adaptive gradient methods in training large-scale neural networks (Peng et al., 2024; Liang et al., 2024; Liu et al., 2024b; Jordan et al., 2024; Yuan et al., 2025; Vyas et al., 2025; Pooladzandi & Li, 2024; Li, 2022; Pethick et al., 2025; Xie et al., 2024). A noteworthy example is the LION optimizer (Chen et al., 2023), which was discovered through symbolic search and has demonstrated competitive empirical performance across diverse tasks despite its simple update rule. A theoretical foundation for LION was established through the LION-$\mathcal{K}$ framework (Chen et al., 2024), which generalizes LION and unifies powerful optimization techniques such as mirror descent (Krichene et al., 2015; Beck & Teboulle, 2003), Nesterov momentum (Sun et al., 2023; Nesterov, 1983), Hamiltonian descent (Maddison et al., 2018), Frank–Wolfe algorithms (Pethick et al., 2025; Jaggi, 2013), and decoupled weight decay (Loshchilov & Hutter, 2019; Liu et al., 2025).

The recently proposed MUON optimizer (Jordan et al., 2024) is another compelling development among emerging optimizers. MUON introduces orthogonalized gradient momentum updates via Newton-Schulz iteration (Bernstein & Newhouse, 2024), demonstrating promising empirical results and potential for efficient large-scale model training (Liu et al., 2025). However, its theoretical underpinnings and connections to broader optimization techniques remain unclear.

In this paper, we bridge this gap by embedding MUON within the LION-$\mathcal{K}$ framework, providing not only a theoretical explanation for MUON's empirical success but also a unified perspective that enables natural generalizations and directions for future work. Specifically, we make the following key contributions, which are further outlined in Section 3.

**Muon as a Lion-$\mathcal{K}$ optimizer.** We interpret MUON as a special case of the LION-$\mathcal{K}$ family of optimizers (generalized to matrix-valued parameters) when the convex function $\mathcal{K}$ and its subgradient $\nabla\mathcal{K}$ are chosen as the nuclear norm and matrix sign function, respectively. By situating MUON within the broader LION-$\mathcal{K}$ framework, we naturally enable generalizations beyond the nuclear norm, introducing a richer family of optimizers with various implicit regularization effects derived from alternative convex functions.

**Implicit constrained optimization via Muon.** Under standard assumptions, we show that MUON with decoupled weight decay (Loshchilov & Hutter, 2019; Liu et al., 2025) implicitly solves an optimization problem that enforces spectral norm constraints by converging to its set of Karush–Kuhn–Tucker (KKT) points (Theorems 3 to 6). Theorems 5 and 6 notably provide convergence guarantees in the nonconstant, Robbins–Monro-style step size regime, which has not been explored in prior work.

**Convergence rates of Muon.** We introduce the KKT score (Definition 3), a stationarity measure for the aforementioned constrained optimization problem, and provide convergence rates for the KKT score in both the deterministic (Theorem 3) and stochastic gradient (Theorem 4) settings. In particular, Theorem 4 improves upon Theorem B.14 in Chen et al. (2024) by removing the assumption that $\mathcal{K}$ has a Lipschitz weak gradient, which does not apply to MUON.

## 2 Related work

**Steepest descent under norm constraints.** Bernstein & Newhouse (2024) reinterprets popular optimizers such as ADAM (Kingma & Ba, 2015), SHAMPOO (Gupta et al., 2018), and MUON (Jordan et al., 2024) as instances of steepest descent under norm constraints. Similarly, Pethick et al. (2025) proposes a stochastic conditional gradient approach from a norm-constraint perspective. However, these analyses do not account for momentum, a central component in practical implementations of these optimizers. As a result, when momentum is introduced, these methods no longer strictly conform to the steepest descent interpretation, highlighting a fundamental limitation of this perspective. Moreover, this interpretation does not naturally extend to optimizers that incorporate decoupled weight decay.

**Decoupled weight decay.** Weight decay is a widely used regularization technique in deep learning, traditionally implemented as an $\ell_2$ penalty directly coupled with gradient-based parameter updates (Loshchilov & Hutter, 2019). The concept of decoupled weight decay, introduced by ADAMW (Loshchilov & Hutter, 2019), separates regularization from adaptive gradient computations. Empirical evidence suggests that this decoupling enhances training stability and improves generalization, making it a standard practice in modern adaptive optimizers (Chen et al., 2023; Liu et al., 2025). Recently, Xie & Li (2024) demonstrated that ADAMW implicitly solves a constrained optimization problem given convergence. Chen et al. (2024) proved that optimizers with bounded updates and decoupled weight decay inherently correspond to constrained optimization formulations, even without requiring convergence assumptions.

**Lyapunov analysis of optimizers.** Hamiltonian dynamics provides a rigorous theoretical framework for understanding momentum-based optimization (Nesterov, 1983; Sutskever et al., 2013). Unlike standard gradient descent, which ensures a monotonic decrease in the objective function, momentum methods exhibit nonmonotonic behavior, requiring more advanced analytical tools for convergence analysis (Jin et al.,

2018). Lyapunov-based techniques (Shevitz & Paden, 1994; Krichene et al., 2015; Liu et al., 2024a; Chen et al., 2024) have since been developed to analyze the stability and convergence properties of optimization algorithms (Liang et al., 2024).

**Concurrent work.** Although several concurrent works have studied the convergence of MUON under various smoothness assumptions (Li & Hong, 2025; An et al., 2025; Kovalev, 2025; Shen et al., 2025), none of them consider MUON with decoupled weight decay, despite it being the variant that has demonstrated the most promising empirical results (Liu et al., 2025). Sfyraki & Wang (2025) analyzes MUON with decoupled weight decay through a Frank–Wolfe perspective, whereas our work is the first to use the LION-$\mathcal{K}$ framework and to prove convergence with decreasing step sizes à la Robbins–Monro.

## 3 Main results

In this paper, we consider the optimization problem

$$\min_{\mathbf{X} \in \mathbb{X}} \mathcal{F}(\mathbf{X}) \quad \text{with} \quad \mathcal{F}(\mathbf{X}) = \mathbb{E}_{\xi \sim \mathcal{D}} \left[ \mathcal{F}(\mathbf{X}, \xi) \right], \tag{1}$$

where $\mathbb{X} := \mathbb{R}^{n \times m}$ is the space of real $n \times m$ matrices, $\mathcal{F} : \mathbb{X} \to \mathbb{R}$ is a differentiable loss function, and the expectation is taken over the data distribution $\mathcal{D}$ with independent and identically distributed samples $\xi \sim \mathcal{D}$. Given a realization of the function $\mathcal{F}(\mathbf{X}, \xi)$, the stochastic gradient $\nabla \mathcal{F}(\mathbf{X}, \xi)$ is defined as the gradient of $\mathcal{F}(\mathbf{X}, \xi)$ with respect to the variable $\mathbf{X}$.

The MUON optimizer (Jordan et al., 2024) was recently proposed for solving (1). When equipped with Nesterov momentum and decoupled weight decay (Liu et al., 2025), it has the implicit update rule

$$\begin{aligned}
\mathbf{M}_{t+1} &= \beta_2 \mathbf{M}_t - (1 - \beta_2) \mathbf{G}_t \\
\widetilde{\mathbf{M}}_{t+1} &= \beta_1 \mathbf{M}_t - (1 - \beta_1) \mathbf{G}_t \\
\mathbf{X}_{t+1} &= \mathbf{X}_t + \eta_t \left( \mathrm{msgn}(\widetilde{\mathbf{M}}_{t+1}) - \lambda \mathbf{X}_{t+1} \right),
\end{aligned} \tag{2}$$

where $\mathbf{X}_t$ represents the parameters, $\mathbf{M}_t$ and $\widetilde{\mathbf{M}}_t$ represent momentum, $\mathbf{G}_t$ is either the deterministic gradient $\nabla \mathcal{F}(\mathbf{X}_t)$ or a stochastic gradient $\nabla \mathcal{F}(\mathbf{X}_t, \xi_t)$, $\eta_t > 0$ is the learning rate, $\beta_1, \beta_2 \in [0, 1)$ are two momentum coefficients, $\lambda \geq 0$ is the weight decay coefficient, and msgn is known as the matrix sign function (see Definition 2).

LION-$\mathcal{K}$ (Chen et al., 2024) is a family of optimizers originally developed as a generalization and theoretical foundation for the LION optimizer (Chen et al., 2023). It is parameterized by a convex function $\mathcal{K} : \mathbb{X} \to \mathbb{R}$ with a subgradient $\nabla \mathcal{K}$ and has the implicit update rule

$$\begin{aligned}
\mathbf{M}_{t+1} &= \beta_2 \mathbf{M}_t - (1 - \beta_2) \mathbf{G}_t \\
\widetilde{\mathbf{M}}_{t+1} &= \beta_1 \mathbf{M}_t - (1 - \beta_1) \mathbf{G}_t \\
\mathbf{X}_{t+1} &= \mathbf{X}_t + \eta_t \left( \nabla \mathcal{K}(\widetilde{\mathbf{M}}_{t+1}) - \lambda \mathbf{X}_{t+1} \right).
\end{aligned} \tag{3}$$

The update rule of MUON bears remarkable similarity to (3), and MUON can in fact be identified as the special case of LION-$\mathcal{K}$ with $\mathcal{K}(\mathbf{X}) = \|\mathbf{X}\|_{\mathrm{tr}}$ and $\nabla \mathcal{K}(\mathbf{X}) = \mathrm{msgn}(\mathbf{X})$, where $\|\cdot\|_{\mathrm{tr}}$ denotes the nuclear norm and msgn is known to be a subgradient of $\|\cdot\|_{\mathrm{tr}}$. Recall that $\|\mathbf{X}\|_{\mathrm{tr}} = \sum_{i=1}^{\min(n,m)} \boldsymbol{\sigma}_i(\mathbf{X})$, where $\boldsymbol{\sigma}_i(\mathbf{X})$ is the $i^{\text{th}}$ largest singular value of $\mathbf{X}$.

Perhaps surprisingly, due to decoupled weight decay, LION-$\mathcal{K}$ optimizers do not minimize the original loss function. Instead, they minimize the regularized objective

$$\widehat{\mathcal{F}}(\mathbf{X}) := \mathcal{F}(\mathbf{X}) + \frac{1}{\lambda} \mathcal{K}^*(\lambda \mathbf{X}), \tag{4}$$

where $\mathcal{K}^*$ denotes the convex conjugate of $\mathcal{K}$. Leveraging this property of LION-$\mathcal{K}$, we conclude that MUON is implicitly solving the constrained optimization problem

$$\min_{\mathbf{X} \in \mathbb{X}} \mathcal{F}(\mathbf{X}) \text{ s.t. } \|\mathbf{X}\|_{\mathrm{op}} \leq \frac{1}{\lambda}, \tag{5}$$

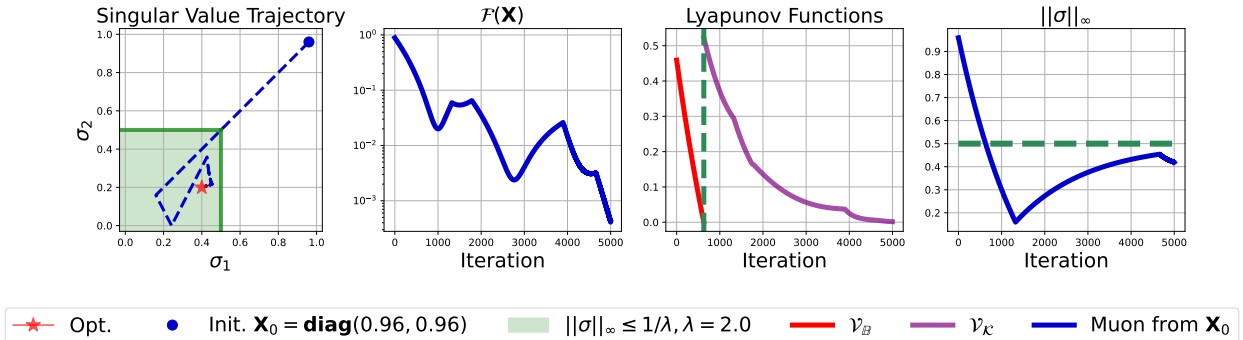

Figure 1: Convergence behavior of MUON. Although the primary objective value $\mathcal{F}(\mathbf{X})$ exhibits nonmonotonic fluctuations, the Lyapnuov functions $\mathcal{V}_{\mathbb{B}}$ and $\mathcal{V}_{\mathcal{K}}$ decrease monotonically within their respective domains — $\mathcal{V}_{\mathbb{B}}$ when the trajectory is outside $\mathbb{B}$, and $\mathcal{V}_{\mathcal{K}}$ once the trajectory enters $\mathbb{B}$.

where the spectral norm $\|\cdot\|_{\mathrm{op}}$, defined as $\|\mathbf{X}\|_{\mathrm{op}} = \boldsymbol{\sigma}_1(\mathbf{X})$, is known to be the dual norm of $\|\cdot\|_{\mathrm{tr}}$.

Despite the general LION-$\mathcal{K}$ framework, MUON's use of the nondifferentiable nuclear norm casts unique challenges in providing convergence guarantees. In this work, we provide an analysis tailored to MUON and rigorously establish that the iterates of (2) converge to the set of KKT points of (5).

To give a quick overview of the results, we first note that the KKT points of (5) can be characterized by the KKT score function

$$\mathcal{S}(\mathbf{X}) := \|\nabla\mathcal{F}(\mathbf{X})\|_{\mathrm{tr}} + \langle \lambda\mathbf{X}, \nabla\mathcal{F}(\mathbf{X}) \rangle. \tag{6}$$

We show in Proposition 2 that a point $\mathbf{X}$ is a KKT point if and only if the KKT score is zero, i.e., $\mathcal{S}(\mathbf{X}) = 0$, and the primal constraint $\|\mathbf{X}\|_{\mathrm{op}} \leq \frac{1}{\lambda}$ is satisfied. We then identify two Lyapunov functions that are used to verify convergence in terms of these conditions. For the constraint condition $\|\mathbf{X}\|_{\mathrm{op}} \leq \frac{1}{\lambda}$, we use the Lyapunov function

$$\mathcal{V}_{\mathbb{B}}(\mathbf{X}) = \max\left( \|\mathbf{X}\|_{\mathrm{op}} - \frac{1}{\lambda}, 0 \right), \tag{7}$$

which measures the distance from $\mathbf{X}$ to the constraint ball $\mathbb{B} := \{ \mathbf{X} \in \mathbb{X} \mid \|\lambda\mathbf{X}\|_{\mathrm{op}} \leq 1 \}$. Following the update (2), we show that $\mathcal{V}_{\mathbb{B}}(\mathbf{X}_t)$ decays exponentially fast when $\eta_t < \frac{1}{\lambda}$, i.e.

$$\mathcal{V}_{\mathbb{B}}(\mathbf{X}_t) \leq \left( \prod_{s=0}^{t-1}(1 - \eta_s\lambda) \right) \mathcal{V}_{\mathbb{B}}(\mathbf{X}_0).$$

Hence, $\mathcal{V}_{\mathbb{B}}$ converges to 0 at a linear rate, which implies that $\mathbf{X}_t$ rapidly converges to $\mathbb{B}$ and never leaves it after entering. Inside the ball, we use a second Lyapunov function

$$\mathcal{V}_{\mathcal{K}}(\mathbf{X}, \mathbf{M}) = \mathcal{F}(\mathbf{X}) - \mathcal{F}^\star + \frac{c}{\lambda}(\|\mathbf{M}\|_{\mathrm{tr}} - \langle \lambda\mathbf{X}, \mathbf{M} \rangle),$$

where $c$ is an appropriately defined scalar. We show that MUON (approximately) monotonically decreases $\mathcal{V}_{\mathcal{K}}$ within the constraint set, which implies that the KKT score vanishes along the trajectory by a generalization of LaSalle's invariance principle (LaSalle, 1960) for discrete-time stochastic processes.

Our main results are summarized by Figure 1 and the following theorems.

**Theorem 1** (Informal, see Theorems 3 and 4)**.** *When $\mathbf{X}_t$, $\mathbf{M}_t$, and $\widetilde{\mathbf{M}}_t$ are updated using (2) with $\eta_t = \eta = \Theta\left(\frac{1}{\sqrt{T}}\right)$ and $\mathrm{Var}(\mathbf{G}_t) \leq \frac{\sigma^2}{n_{\mathrm{batch}}}$, we have that $\frac{1}{T}\sum_{t=1}^{T}\mathbb{E}[\mathcal{S}(\mathbf{X}_t)] = O\left(\frac{1}{\sqrt{T}} + \frac{\sigma}{\sqrt{n_{\mathrm{batch}}}}\right)$.*

**Theorem 2** (Informal, see Theorems 5 and 6)**.** *Assume $\eta_t \leq \frac{1}{\lambda}$, $\sum_{t=0}^{\infty}\eta_t = \infty$, and $\sum_{t=0}^{\infty}\eta_t^2 < \infty$. Under certain conditions, when $\mathbf{X}_t$, $\mathbf{M}_t$, and $\widetilde{\mathbf{M}}_t$ are updated using (2), we have that $\mathbf{X}_t$ converges to the set of KKT points of (5) a.s., regardless of initialization.*

Precise statements and detailed proofs can be found in Section 7.

## 4 Preliminaries

**General notation.** We let $\mathbb{X} := \mathbb{R}^{n \times m}$ denote the space of real $n \times m$ matrices, corresponding to weight matrices in neural networks. We denote matrices in capital boldface and vectors in lowercase boldface. We let $\mathbf{0}$ denote the zero matrix of appropriate dimension. We let $\langle \mathbf{X}, \mathbf{Y} \rangle := \mathrm{Tr}(\mathbf{X}^\top \mathbf{Y})$ denote the Frobenius inner product. For a differentiable function $\mathcal{K} : \mathbb{X} \to \mathbb{R} \cup \{\infty\}$, we let $\nabla \mathcal{K}$ denote the gradient of $\mathcal{K}$. We say $\mathcal{K}$ is *convex* if for all $\mathbf{X}, \mathbf{Y} \in \mathbb{X}$ and $\lambda \in [0, 1]$,

$$\mathcal{K}((1 - \lambda)\mathbf{X} + \lambda \mathbf{Y}) \leq (1 - \lambda)\mathcal{K}(\mathbf{X}) + \lambda \mathcal{K}(\mathbf{Y}).$$

We say $\mathbf{G}$ is a *subgradient* of $\mathcal{K}$ at $\mathbf{X}$ if for all $\mathbf{Y} \in \mathbb{X}$,

$$\mathcal{K}(\mathbf{Y}) \geq \mathcal{K}(\mathbf{X}) + \langle \mathbf{G}, \mathbf{Y} - \mathbf{X} \rangle.$$

If $\mathcal{K}$ is convex and differentiable at $\mathbf{X}$, then $\nabla \mathcal{K}(\mathbf{X})$ is the unique subgradient of $\mathcal{K}$ at $\mathbf{X}$. If $\mathcal{K}$ is convex but nondifferentiable, we let $\partial \mathcal{K}(\mathbf{X})$ denote the set of subgradients of $\mathcal{K}$ at $\mathbf{X}$ and overload $\nabla \mathcal{K}(\mathbf{X})$ to denote an element of $\partial \mathcal{K}(\mathbf{X})$. For a function $\mathcal{K}$, we let $\mathcal{K}^*$ denote the convex conjugate of $\mathcal{K}$, where

$$\mathcal{K}^*(\mathbf{X}) := \sup_{\mathbf{Y} \in \mathbb{X}} \left( \langle \mathbf{X}, \mathbf{Y} \rangle - \mathcal{K}(\mathbf{Y}) \right).$$

From this definition, we immediately deduce the *Fenchel–Young inequality*

$$\mathcal{K}(\mathbf{X}) + \mathcal{K}^*(\mathbf{Y}) \geq \langle \mathbf{X}, \mathbf{Y} \rangle.$$

We let $\mathrm{dom}(\mathcal{K}) := \{\mathbf{X} \in \mathbb{X} \mid \mathcal{K}(\mathbf{X}) < \infty\}$ denote the effective domain of $\mathcal{K}$. We say $\mathcal{K} : \mathbb{X} \to \mathbb{R} \cup \{\infty\}$ is *closed* if the set $\{\mathbf{X} \in \mathrm{dom}(\mathcal{K}) \mid \mathcal{K}(\mathbf{X}) \leq \alpha\}$ is closed for each $\alpha \in \mathbb{R}$. We say $\mathcal{K} : \mathbb{X} \to \mathbb{R} \cup \{\infty\}$ is *proper* if $\mathrm{dom}(\mathcal{K})$ is nonempty. The celebrated *Fenchel–Moreau theorem* states that if $\mathcal{K} : \mathbb{X} \to \mathbb{R} \cup \{\infty\}$ is convex, closed, and proper, then $\mathcal{K}^{**} = \mathcal{K}$. A corollary is that $\mathbf{Y} \in \partial \mathcal{K}(\mathbf{X})$ if and only if

$$\mathcal{K}(\mathbf{X}) + \mathcal{K}^*(\mathbf{Y}) = \langle \mathbf{X}, \mathbf{Y} \rangle,$$

i.e. the Fenchel–Young inequality holds with equality, and $\mathbf{X} \in \partial \mathcal{K}^*(\mathbf{Y})$ if $\partial \mathcal{K}^*(\mathbf{Y})$ is nonempty. We let $\chi_\mathbb{D}$ denote the characteristic function of a set $\mathbb{D} \subseteq \mathbb{X}$, where

$$\chi_\mathbb{D}(\mathbf{X}) := \begin{cases} 0 & \text{if } \mathbf{X} \in \mathbb{D} \\ \infty & \text{otherwise} \end{cases}.$$

For $k \in \{1, 2, \ldots, \min(n, m)\}$, we let $\boldsymbol{\sigma}_k(\mathbf{X})$ denote the $k^{\text{th}}$ largest singular value of $\mathbf{X}$.

**Norms.** For a norm $\|\cdot\|$ on $\mathbb{X}$, and $r > 0$, we let $\mathbb{B}_{\|\cdot\|}(r) := \{\mathbf{X} \in \mathbb{X} \mid \|\mathbf{X}\| \leq r\}$ denote the ball of radius $r$. When $\mathbb{D} \subseteq \mathbb{X}$ and the norm is clear from context, we let $d(\mathbf{X}, \mathbb{D}) := \inf_{\mathbf{Y} \in \mathbb{D}} \|\mathbf{X} - \mathbf{Y}\|$ denote the distance from $\mathbf{X}$ to $\mathbb{D}$. We let $\|\cdot\|_*$ denote the dual norm of $\|\cdot\|$, where

$$\|\mathbf{X}\|_* := \sup_{\mathbf{Y} \neq \mathbf{0}} \frac{\langle \mathbf{X}, \mathbf{Y} \rangle}{\|\mathbf{Y}\|}.$$

It follows directly from this definition that $\langle \mathbf{X}, \mathbf{Y} \rangle \leq \|\mathbf{X}\| \|\mathbf{Y}\|_*$ and $\|\mathbf{X}\|_{**} = \|\mathbf{X}\|$. We define the matrix norms

$$\|\mathbf{X}\|_p := \left( \sum_{i=1}^n \sum_{j=1}^m |\mathbf{X}_{ij}|^p \right)^{\frac{1}{p}}, \ \|\mathbf{X}\|_{\mathrm{tr}} := \sum_{i=1}^{\min(n,m)} \boldsymbol{\sigma}_i(\mathbf{X}), \|\mathbf{X}\|_{\mathrm{F}} := \|\mathbf{X}\|_2, \|\mathbf{X}\|_{\mathrm{op}} := \boldsymbol{\sigma}_1(\mathbf{X}),$$

where $p \in [1, \infty]$. $\|\cdot\|_p$ is the entrywise $\ell_p$ norm, $\|\cdot\|_{\mathrm{tr}}$ is known as the *trace norm* or *nuclear norm*, $\|\cdot\|_{\mathrm{F}}$ is known as the *Frobenius norm*, and $\|\cdot\|_{\mathrm{op}}$ is known as the *spectral norm*. The dual norm of $\|\cdot\|_p$ is $\|\cdot\|_q$, where $\frac{1}{p} + \frac{1}{q} = 1$, the dual norm of $\|\cdot\|_{\mathrm{tr}}$ is $\|\cdot\|_{\mathrm{op}}$, and $\|\cdot\|_{\mathrm{F}}$ is self-dual.

**Fact 1.** *Let $\|\cdot\|$ be a norm on $\mathbb{X}$ with dual norm $\|\cdot\|_*$. If $\mathcal{K}(\mathbf{X}) = \|\mathbf{X}\|$, then*

$$\mathcal{K}^*(\mathbf{X}) = \chi_{\mathbb{B}_*(1)}(\mathbf{X}) = \begin{cases} 0 & \text{if } \|\mathbf{X}\|_* \leq 1 \\ \infty & \text{otherwise} \end{cases}.$$

We say that a function $\mathcal{F} : \mathbb{X} \to \mathbb{R}$ is *L-smooth* if it is differentiable and

$$\|\nabla\mathcal{F}(\mathbf{Y}) - \nabla\mathcal{F}(\mathbf{X})\|_{\mathrm{F}} \leq L \|\mathbf{Y} - \mathbf{X}\|_{\mathrm{F}} \text{ for all } \mathbf{X}, \mathbf{Y} \in \mathbb{X}.$$

If $\mathcal{F}$ is $L$-smooth, then

$$\mathcal{F}(\mathbf{Y}) \leq \mathcal{F}(\mathbf{X}) + \langle \nabla\mathcal{F}(\mathbf{X}), \mathbf{Y} - \mathbf{X} \rangle + \frac{L}{2} \|\mathbf{Y} - \mathbf{X}\|_{\mathrm{F}}^2 \text{ for all } \mathbf{X}, \mathbf{Y} \in \mathbb{X}.$$

For additional background on convex analysis, we refer to Rockafellar (1970).

## 4.1 Assumptions

**Assumption 1.** $\mathcal{F}^\star := \inf_{\mathbf{X} \in \mathbb{X}} \mathcal{F}(\mathbf{X})$ *is finite, and there exists $\mathbf{X}^\star \in \mathbb{X}$ such that $\mathcal{F}(\mathbf{X}^\star) = \mathcal{F}^\star$.*

Assumption 1 is necessary for (1) to be well-posed. For our discrete-time analysis, we impose an additional smoothness assumption on $\mathcal{F}$.

**Assumption 2** (*L*-smoothness). $\mathcal{F} : \mathbb{X} \to \mathbb{R}$ *is L-smooth.*

We now define the variance of random matrices and introduce an assumption for the analysis of stochastic settings.

**Definition 1** (Variance). *The* variance *of an $\mathbb{X}$-valued random variable $\mathbf{X}$ is defined as*

$$\mathrm{Var}(\mathbf{X}) := \mathbb{E}\left[\|\mathbf{X} - \mathbb{E}[\mathbf{X}]\|_{\mathrm{F}}^2\right].$$

**Assumption 3** (Bounded variance). *The stochastic samples $\xi_t \sim \mathcal{D}$ are independent and identically distributed (i.i.d.). Additionally, the stochastic gradient $\nabla\mathcal{F}(\mathbf{X}_t, \xi_t)$ satisfies*

$$\mathbb{E}_{\xi_t \sim \mathcal{D}}[\nabla\mathcal{F}(\mathbf{X}_t, \xi_t)] = \nabla\mathcal{F}(\mathbf{X}_t) \quad \text{and} \quad \mathrm{Var}(\nabla\mathcal{F}(\mathbf{X}_t, \xi_t)) \leq \frac{\sigma^2}{n_{\mathrm{batch}}},$$

*where $\sigma^2$ is a constant and $n_{\mathrm{batch}}$ denotes the batch size.*

**Assumption 4** (Iteration-wise bounded variance). *The stochastic gradient $\nabla\mathcal{F}(\mathbf{X}_t, \xi_t)$ satisfies*

$$\mathbb{E}_{\xi_t \sim \mathcal{D}}[\nabla\mathcal{F}(\mathbf{X}_t, \xi_t)] = \nabla\mathcal{F}(\mathbf{X}_t) \quad \text{and} \quad \mathrm{Var}(\nabla\mathcal{F}(\mathbf{X}_t, \xi_t)) \leq \left[\frac{\sigma^2}{n_{\mathrm{batch}}}\right]_t.$$

We remark that Assumptions 2 and 3 are standard in the literature for the analysis of stochastic optimization algorithms, e.g. Bernstein et al. (2018); Défossez et al. (2022); Liu et al. (2024a); Yuan et al. (2025).

Since we work within the LION-$\mathcal{K}$ framework, our last assumption concerns the choice of $\mathcal{K}$.

**Assumption 5.** $\mathcal{K} : \mathbb{X} \to \mathbb{R}$ *is convex. This implies that $\mathcal{K}$ is also closed and proper.*

## 5 Background on Lion-$\mathcal{K}$

LION-$\mathcal{K}$ (Chen et al., 2024) is a family of optimization algorithms developed to provide a theoretical foundation for the LION optimizer, which was originally discovered via symbolic search (Chen et al., 2023). Given a convex function $\mathcal{K} : \mathbb{X} \to \mathbb{R}$ with subgradient $\nabla\mathcal{K}$, the update rule for LION-$\mathcal{K}$ is given by (3). This update rule is equivalent to the original one given by Chen et al. (2024), where the last update is

$$\mathbf{X}_{t+1} = \mathbf{X}_t + \eta_t(\nabla\mathcal{K}(\widetilde{\mathbf{M}}_{t+1}) - \lambda\mathbf{X}_t), \tag{8}$$

under the reparameterization $\eta_t \leftarrow \frac{\eta_t}{1+\eta_t \lambda}$.

LION-$\mathcal{K}$ can be seen as mixing several fundamental design elements in optimization:

- **Polyak momentum M**, which accumulates the exponential moving average of the gradients, controlled by the coefficient $\beta_2$.

- **Nesterov momentum $\widetilde{\mathbf{M}}$**, which introduces extra gradient components into the update, controlled by the coefficient $\beta_1$ (see Appendix B.2, Chen et al. (2024)).

- **Nonlinear preconditioning $\nabla \mathcal{K}$**, which applies a transformation to the momentum before it is used to update the parameters. This is legitimate since $\nabla \mathcal{K}$ is a monotone map, meaning that $\langle \nabla \mathcal{K}(\mathbf{X}) - \nabla \mathcal{K}(\mathbf{Y}), \mathbf{X} - \mathbf{Y} \rangle \geq 0$, which follows from the convexity of $\mathcal{K}$ (see Lemma 1).

- **Decoupled weight decay $\lambda \mathbf{X}$**, which reduces the parameter magnitude in addition to the update $\nabla \mathcal{K}(\widetilde{\mathbf{M}})$. This introduces a regularization effect (see Section 5.1) and is closely related to Frank–Wolfe style algorithms.

### 5.1  Effect of decoupled weight decay

Due to the interplay of decoupled weight decay and the $\nabla \mathcal{K}$ mapping, LION-$\mathcal{K}$ optimizers minimize the regularized objective (4). To gain a quick heuristic understanding of how the regularization term arises, we can simply examine a fixed point of the optimizer. Using the original update rule (8), assume that the algorithm reaches a fixed point, where we have $\mathbf{M}_{t+1} = \widetilde{\mathbf{M}}_{t+1} = -\nabla \mathcal{F}(\mathbf{X}_t)$ and $\nabla \mathcal{K}(\widetilde{\mathbf{M}}_{t+1}) - \lambda \mathbf{X}_t = \mathbf{0}$. This yields $\nabla \mathcal{K}(-\nabla \mathcal{F}(\mathbf{X}_t)) - \lambda \mathbf{X}_t = \mathbf{0}$. Since $\nabla \mathcal{K}^*$ is the inverse function of $\nabla \mathcal{K}$ by convex conjugacy, we have

$$\nabla \widehat{\mathcal{F}}(\mathbf{X}_t) = \nabla \mathcal{F}(\mathbf{X}_t) + \nabla \mathcal{K}^*(\lambda \mathbf{X}_t) = \mathbf{0}.$$

This suggests that every fixed point of the algorithm must be a stationary point of the regularized objective $\widehat{\mathcal{F}}$.

### 5.2  Lyapunov function for Lion-$\mathcal{K}$

The fixed-point analysis alone does not guarantee the convergence of the algorithm. We give a full analysis using a Lyapunov function method, patterned off Chen et al. (2024). To understand this, it helps to focus on the limit of small step sizes, where the dynamics of LION-$\mathcal{K}$ can be modeled by the ordinary differential equation (ODE)

$$\begin{aligned}
\dot{\mathbf{M}}_t &= -\nabla \mathcal{F}(\mathbf{X}_t) - \mathbf{M}_t \\
\dot{\mathbf{X}}_t &= \nabla \mathcal{K}\left(\mathbf{M}_t - \epsilon\left(\nabla \mathcal{F}(\mathbf{X}_t) + \mathbf{M}_t\right)\right) - \lambda \mathbf{X}_t.
\end{aligned} \tag{9}$$

Here, the effect of Nesterov momentum is captured by $\epsilon \in [0, 1]$.

It is not immediately obvious why the LION-$\mathcal{K}$ ODE would serve to minimize $\widehat{\mathcal{F}}(\mathbf{X})$, as the ODE does not necessarily guarantee a monotonic decrease in $\widehat{\mathcal{F}}(\mathbf{X})$. However, we show in Appendix A that the LION-$\mathcal{K}$ ODE minimizes the auxiliary function

$$\mathcal{H}(\mathbf{X}, \mathbf{M}) := \widehat{\mathcal{F}}(\mathbf{X}) + \frac{1-\epsilon}{1+\epsilon\lambda}\left(\mathcal{K}^*(\lambda \mathbf{X}) + \mathcal{K}(\mathbf{M}) - \langle \mathbf{M}, \lambda \mathbf{X} \rangle\right)$$

in the sense that it is monotonically decreasing along the ODE trajectories, i.e. $\frac{\mathrm{d}}{\mathrm{d}t}\mathcal{H}(\mathbf{X}_t, \mathbf{M}_t) \leq 0$ until a local minimum is achieved. In other words, $\mathcal{H}(\mathbf{X}, \mathbf{M})$ admits a Lyapunov function of the LION-$\mathcal{K}$ ODE (9). Here, $\mathcal{H}(\mathbf{X}, \mathbf{M})$ is a joint function of position $\mathbf{X}$ and momentum $\mathbf{M}$. It can be interpreted as a Hamiltonian function in a physical metaphor, where $\widehat{\mathcal{F}}(\mathbf{X})$ is the potential energy, and the additional term represents a form of kinetic energy.

Moreover, minimizing $\mathcal{H}(\mathbf{X}, \mathbf{M})$ is equivalent to minimizing the regularized objective $\widehat{\mathcal{F}}(\mathbf{X})$. This can be seen by the Fenchel–Young inequality, which ensures that $\mathcal{K}^*(\lambda \mathbf{X}) + \mathcal{K}(\mathbf{M}) - \langle \mathbf{M}, \lambda \mathbf{X} \rangle \geq 0$, with equality when $\mathbf{M} \in \partial \mathcal{K}(\lambda \mathbf{X})$.

### 5.3 Constrained optimization problem

If $\mathcal{K}^*$ takes on positive infinite values, LION-$\mathcal{K}$ effectively solves the constrained optimization problem

$$\min_{\mathbf{X} \in \mathbb{X}} \widehat{\mathcal{F}}(\mathbf{X}) \text{ s.t. } \lambda \mathbf{X} \in \text{dom}(\mathcal{K}^*). \tag{10}$$

If the algorithm is initialized outside the effective domain, Chen et al. (2024) shows that in the continuous-time setting, $\mathbf{X}_t$ is rapidly driven into the effective domain and stays inside afterwards. Specifically, this process is guaranteed to be exponentially fast in time:

$$d(\lambda \mathbf{X}_t, \text{dom}(\mathcal{K}^*)) \leq \exp(-\lambda t) d(\lambda \mathbf{X}_0, \text{dom}(\mathcal{K}^*)) \text{ for all } t \geq 0. \tag{11}$$

Consequently, $\lambda \mathbf{X}_t$ rapidly converges to $\text{dom}(\mathcal{K}^*)$ and remains within this domain once it arrives, where the Lyapunov function is finite and decreases monotonically.

## 6 Muon meets Lion-$\mathcal{K}$

We recall the MUON update rule (2) and formally define the matrix sign function.

**Definition 2** (Matrix sign). *Let $\mathbf{U\Sigma V}^\top$ be a singular value decomposition of $\mathbf{X} \in \mathbb{X}$. The matrix sign function, denoted* msgn, *is given by*

$$\text{msgn}(\mathbf{X}) := (\mathbf{X}\mathbf{X}^\top)^{-\frac{1}{2}}\mathbf{X} = \mathbf{U}\,\text{sgn}(\mathbf{\Sigma})\mathbf{V}^\top,$$

*where $(\cdot)^{-\frac{1}{2}}$ denotes the Moore–Penrose inverse of the matrix square root and* sgn *denotes the entrywise signum function.*

The matrix sign of $\mathbf{X}$ is also known as the *Mahalanobis whitening* or *zero-phase component analysis (ZCA) whitening*, which stands as the optimal whitening procedure that minimizes the distortion with the original data. It is also closely related to the polar decomposition of $\mathbf{X}$.

We now illustrate the connection between MUON and LION-$\mathcal{K}$ using the following well-known fact.

**Fact 2** (Watson (1992); Candès & Recht (2009)). *Let $\mathcal{K}(\mathbf{X}) = \|\mathbf{X}\|_{\text{tr}}$ and $\mathbf{X} \in \mathbb{X}$ with singular value decomposition $\mathbf{U\Sigma V}^\top$. Then*

$$\partial\mathcal{K}(\mathbf{X}) = \left\{\mathbf{U}\,\text{sgn}(\mathbf{\Sigma})\mathbf{V}^\top + \mathbf{W} \mid \mathbf{W} \in \mathbb{X}, \mathbf{U}^\top\mathbf{W} = \mathbf{0}, \mathbf{WV} = \mathbf{0}, \|\mathbf{W}\|_{\text{op}} \leq 1\right\}.$$

*In particular,* msgn$(\mathbf{X}) \in \partial\mathcal{K}(\mathbf{X})$.

Hence, MUON can be interpreted as LION-$\mathcal{K}$ with $\mathcal{K}(\mathbf{X}) = \|\mathbf{X}\|_{\text{tr}}$ and $\nabla\mathcal{K}(\mathbf{X}) = \text{msgn}(\mathbf{X})$, corresponding to a matrix generalization of LION, which is LION-$\mathcal{K}$ with $\mathcal{K}(\mathbf{X}) = \|\mathbf{X}\|_1$ and $\nabla\mathcal{K}(\mathbf{X}) = \text{sgn}(\mathbf{X})$. Indeed, $\|\cdot\|_{\text{tr}}$ and $\|\cdot\|_{\text{op}}$ are precisely the Schatten 1- and $\infty$-norms, which are the $\ell_1$ and $\ell_\infty$ norms on the singular values of a matrix, respectively, and the suggestively named msgn function can be seen as a matrix analog of sgn.

### 6.1 Spectral norm constraint

By (10) and Fact 1, we conclude that MUON solves the constrained optimization problem (5). The bound $\frac{1}{\lambda}$ is determined solely by the weight decay coefficient $\lambda$. Without weight decay ($\lambda = 0$), we obtain the original unconstrained optimization problem.

In fact, the bound constraint arises from any update of the form

$$\mathbf{X}_{t+1} = \mathbf{X}_t + \eta_t(\mathbf{O}_t - \lambda\mathbf{X}_t),$$

where $\mathbf{O}_t$ has bounded norm, regardless of how it is updated, although the solution may not necessarily minimize the objective within the constrained set. Because $\mathbf{O}_t$ has bounded norm, the weight decay term dominates the update whenever the constraint is not satisfied (i.e. $\|\lambda\mathbf{X}_t\| > 1$), leading to an exponential decrease in magnitude. This intuition is formalized by the following result, which is a discrete-time variant of (11).

**Proposition 1.** *For any update of the form* $\mathbf{X}_{t+1} = \mathbf{X}_t + \eta_t(\mathbf{O}_t - \lambda \mathbf{X}_t)$ *with* $\|\mathbf{O}_t\| \leq b$, $\eta_t \lambda \leq 1$, *and* $\lambda > 0$, *where* $\|\cdot\|$ *is any norm on* $\mathbb{X}$ *and* $b$ *is a constant, we have*

$$\|\mathbf{X}_t\| - \frac{b}{\lambda} \leq \left(\prod_{s=0}^{t-1}(1 - \eta_s \lambda)\right)\left(\|\mathbf{X}_0\| - \frac{b}{\lambda}\right).$$

*Proof.* We have

$$\|\mathbf{X}_{t+1}\| - \frac{b}{\lambda} = \|\mathbf{X}_t + \eta_t(\mathbf{O}_t - \lambda \mathbf{X}_t)\| - \frac{b}{\lambda} \leq (1 - \eta_t \lambda)\|\mathbf{X}_t\| + \eta_t \|\mathbf{O}_t\| - \frac{b}{\lambda}$$
$$\leq (1 - \eta_t \lambda)\|\mathbf{X}_t\| + \eta_t b - \frac{b}{\lambda} = (1 - \eta_t \lambda)\left(\|\mathbf{X}_t\| - \frac{b}{\lambda}\right).$$

Applying this recursively yields the result. □

For MUON, we have $\|\mathrm{msgn}(\widetilde{\mathbf{M}}_{t+1})\|_{\mathrm{op}} \leq 1$, so Proposition 1 applies.

Although the continuous-time results in Section 5 provide intuition on the dynamics of LION-$\mathcal{K}$, the non-differentiability of the trace norm ultimately prevents us from directly applying these results in establishing the theoretical properties of MUON. Instead, we will resort to our discrete-time analysis in Section 7 to rigorously prove the convergence and implicit bias of MUON.

## 6.2 Generalizations of Muon

Generalizing beyond the nuclear norm, we can take $\mathcal{K}$ to be a general convex spectral function, i.e.

$$\mathcal{K}(\mathbf{X}) = \sum_{i=1}^{\min(n,m)} \phi(\boldsymbol{\sigma}_i(\mathbf{X})),$$

where $\phi : [0, \infty) \to \mathbb{R}$ is a convex scalar function. Because $\frac{\partial \boldsymbol{\sigma}_i(\mathbf{X})}{\partial \mathbf{X}} = \mathbf{u}_i \mathbf{v}_i^\top$, where $\mathbf{u}_i$ and $\mathbf{v}_i$ are the singular vectors associated with $\boldsymbol{\sigma}_i$, a subgradient of $\mathcal{K}$ above is given by

$$\nabla \mathcal{K}(\mathbf{X}) = \mathbf{U}\mathbf{diag}\left(\{\nabla\phi(\boldsymbol{\sigma}_i)\}\right)\mathbf{V}^\top,$$

where $\mathbf{X}$ has singular value decomposition $\mathbf{U}\boldsymbol{\Sigma}\mathbf{V}^\top$.

Assume $\nabla\phi$ is upper bounded by $b$, i.e. $\sup_{x \geq 0} \nabla\phi(x) = b$. Then the update $\mathbf{O}_t = \nabla \mathcal{K}(\widetilde{\mathbf{M}}_{t+1})$ satisfies $\|\mathbf{O}_t\|_{\mathrm{op}} \leq b$, which yields a constraint of $\|\mathbf{X}\|_{\mathrm{op}} \leq \frac{b}{\lambda}$ by Proposition 1.

In the practical implementation of MUON, $\phi$ is effectively taken as a high-order polynomial inspired by Newton–Schulz iteration for calculating the matrix sign.

Table 1 summarizes several common convex matrix functions along with their key properties. Importantly, the convexity of $\mathcal{K}$ ensures the convergence of the associated LION-$\mathcal{K}$ optimizers, as established through both continuous-time and discrete-time analyses (Section 7 and Appendix A). Consequently, our framework introduces a large class of provably convergent optimization algorithms parameterized by a convex function $\mathcal{K}$.

More generally, the constrained optimization problem $\min_{\mathbf{X} \in \mathbb{X}} \mathcal{F}(\mathbf{X})$ such that $\mathbf{X} \in \mathbb{D}$, where $\mathbb{D}$ is a convex set, can be solved using LION-$\mathcal{K}$ with $\mathcal{K}^*(\mathbf{X}) = \chi_{\mathbb{D}}(\mathbf{X})$. As in the cases of LION and MUON, a particularly important instance is when $\mathbb{D}$ is a norm ball $\mathbb{B}_{\|\cdot\|}(1)$. In this setting, $\nabla \mathcal{K}(\mathbf{X})$ corresponds to the solution of a linear minimization oracle problem $\max_{\mathbf{Z} \in \mathbb{X}} \langle \mathbf{X}, \mathbf{Z} \rangle$ such that $\|\mathbf{Z}\| \leq 1$. Related discussions can be found, for example, in Bernstein & Newhouse (2024) and Pethick et al. (2025).

Table 1: Summary of common convex matrix functions. Note that the nuclear norm is a special case of the spectral sum with $\phi(\cdot) = |\cdot|$.

| Function | Domain | Remarks |
|---|---|---|
| Squared Frobenius norm $\|\mathbf{X}\|_{\mathrm{F}}^2$ | All matrices | Smooth, strongly convex |
| Nuclear norm $\|\mathbf{X}\|_{\mathrm{tr}}$ | All matrices | Lipschitz continuous, convex |
| Spectral norm $\|\mathbf{X}\|_{\mathrm{op}}$ | All matrices | Lipschitz continuous, convex |
| Quadratic form $\mathrm{Tr}(\mathbf{X}^\top \mathbf{M} \mathbf{X})$ | All matrices, $\mathbf{M} \succeq \mathbf{0}$ | Smooth quadratic form, convex |
| Spectral sum $\sum_i \phi(\boldsymbol{\sigma}_i(\mathbf{X}))$ | All matrices, convex $\phi$ | General convex spectral functions |

## 7 Convergence analysis of Muon

In this section, we generalize the analysis of Chen et al. (2024) to handle $\mathbb{X}$-valued updates and leverage this result to provide convergence rates for the KKT score function (6) and prove the convergence of Muon to the set of KKT points of (5). Our strategy consists of three components:

- In Section 7.1, we show that $\mathbf{X}^\star$ is a KKT point of (5) if and only if $\|\lambda \mathbf{X}^\star\|_{\mathrm{op}} \leq 1$ and the KKT score function (6) vanishes at $\mathbf{X}^\star$.

- We give a discrete-time analysis of matrix Lion-$\mathcal{K}$ and show that, as a corollary, the KKT score function is $O\left(\frac{1}{\sqrt{T}}\right)$ in the deterministic gradient setting (Section 7.2) and $O\left(\frac{1}{\sqrt{T}} + \frac{\sigma}{\sqrt{n_{\mathrm{batch}}}}\right)$ in the stochastic gradient setting (Section 7.3).

- In Section 7.4, we put together the previous results and conclude that Muon converges to the set of KKT points of (5) using LaSalle's invariance principle (LaSalle, 1960).

### 7.1 KKT points of spectral-norm-constrained problems

We note that (5) is equivalent to

$$\min_{\mathbf{X} \in \mathbb{X}} \mathcal{F}(\mathbf{X}) \text{ s.t. } \boldsymbol{\sigma}_i(\mathbf{X}) \leq \frac{1}{\lambda} \text{ for all } i \in \{1, 2, \ldots, \min(n, m)\}$$

and define the KKT points of this constrained optimization problem.

**Definition 3** (KKT points). *We say that $\mathbf{X}^\star \in \mathbb{X}$ is a point that satisfies the Karush–Kuhn–Tucker (KKT) conditions, or is a* KKT *point, of* (5) *if there exists $\boldsymbol{\mu} \in \mathbb{R}^{\min(n,m)}$ such that the following conditions hold:*

- *(stationarity)* $\nabla \mathcal{F}(\mathbf{X}^\star) + \sum_{i=1}^{\min(n,m)} \boldsymbol{\mu}_i \mathbf{u}_i \mathbf{v}_i^\top = \mathbf{0}$, *where $\mathbf{u}_i$ and $\mathbf{v}_i$ are singular vectors corresponding to $\boldsymbol{\sigma}_i(\mathbf{X}^\star)$.*

- *(primal feasibility)* $\boldsymbol{\sigma}_i(\mathbf{X}^\star) \leq \frac{1}{\lambda}$ *for all $i \in \{1, 2, \ldots, \min(n, m)\}$.*

- *(dual feasibility)* $\boldsymbol{\mu} \geq \mathbf{0}$ *entrywise.*

- *(complementary slackness)* $\boldsymbol{\mu}_i \left( \boldsymbol{\sigma}_i(\mathbf{X}^\star) - \frac{1}{\lambda} \right) = 0$ *for all $i \in \{1, 2, \ldots, \min(n, m)\}$.*

At a given point $\mathbf{X}$, it is not immediately clear whether the KKT conditions are satisfied. To address this challenge, we work with an equivalent characterization based on the KKT score function (6). We show the direction used in our analysis here and defer the rest of the proof to Appendix B.

**Proposition 2.** $\mathbf{X}^\star \in \mathbb{X}$ *is a KKT point of* (5) *if and only if* $\|\lambda \mathbf{X}^\star\|_{\mathrm{op}} \leq 1$ *and* $\mathcal{S}(\mathbf{X}^\star) = 0$.

*Proof.* Suppose $\|\lambda \mathbf{X}^\star\|_{\mathrm{op}} \leq 1$ and

$$\mathcal{S}(\mathbf{X}^\star) = \|\nabla \mathcal{F}(\mathbf{X}^\star)\|_{\mathrm{tr}} + \langle \lambda \mathbf{X}^\star, \nabla \mathcal{F}(\mathbf{X}^\star) \rangle = \|-\nabla \mathcal{F}(\mathbf{X}^\star)\|_{\mathrm{tr}} - \langle \lambda \mathbf{X}^\star, -\nabla \mathcal{F}(\mathbf{X}^\star) \rangle = 0.$$

Then $\lambda \mathbf{X}^\star$ is a subgradient of $\|\cdot\|_{\mathrm{tr}}$ at $-\nabla \mathcal{F}(\mathbf{X}^\star)$, since for all $\mathbf{Y} \in \mathbb{X}$,

$$\|\mathbf{Y}\|_{\mathrm{tr}} \geq \|\mathbf{Y}\|_{\mathrm{tr}} \|\lambda \mathbf{X}^\star\|_{\mathrm{op}} \geq \langle \lambda \mathbf{X}^\star, \mathbf{Y} \rangle = \|-\nabla \mathcal{F}(\mathbf{X}^\star)\|_{\mathrm{tr}} + \langle \lambda \mathbf{X}^\star, \mathbf{Y} + \nabla \mathcal{F}(\mathbf{X}^\star) \rangle.$$

Let $\mathbf{U}\boldsymbol{\Sigma}\mathbf{V}^\top$ be a singular value decomposition of $-\nabla \mathcal{F}(\mathbf{X}^\star)$. By Fact 2,

$$\lambda \mathbf{X}^\star = \mathbf{U}\operatorname{sgn}(\boldsymbol{\Sigma})\mathbf{V}^\top + \mathbf{W}, \text{ where } \mathbf{U}^\top \mathbf{W} = \mathbf{0}, \ \mathbf{W}\mathbf{V} = \mathbf{0}, \text{ and } \|\mathbf{W}\|_{\mathrm{op}} \leq 1.$$

Then setting $\boldsymbol{\mu}$ to be the singular values of $\nabla \mathcal{F}(\mathbf{X}^\star)$ in nonincreasing order shows that $\mathbf{X}^\star$ satisfies the KKT conditions:

- (stationarity) since $\mathbf{U}^\top \mathbf{W} = \mathbf{0}$ and $\mathbf{W}\mathbf{V} = \mathbf{0}$, $\lambda \mathbf{X}^\star$ has singular value decomposition

$$\begin{pmatrix} \mathbf{u}_1 & \cdots & \mathbf{u}_r & \mathbf{x}_1 & \cdots & \mathbf{x}_{n-r} \end{pmatrix} \mathbf{diag}\left(1, \ldots, 1, \boldsymbol{\sigma}_1(\mathbf{W}), \ldots, \boldsymbol{\sigma}_{\min(n,m)-r}(\mathbf{W})\right) \begin{pmatrix} \mathbf{v}_1^\top \\ \vdots \\ \mathbf{v}_r^\top \\ \mathbf{y}_1^\top \\ \vdots \\ \mathbf{y}_{m-r}^\top \end{pmatrix},$$

  where $r := \operatorname{rank}(\nabla \mathcal{F}(\mathbf{X}^\star))$. In addition, $\|\mathbf{W}\|_{\mathrm{op}} \leq 1$ implies that $\boldsymbol{\sigma}_1(\mathbf{X}^\star) = \cdots = \boldsymbol{\sigma}_r(\mathbf{X}^\star) = \frac{1}{\lambda}$, with corresponding singular vectors $\mathbf{u}_1, \ldots, \mathbf{u}_r$ and $\mathbf{v}_1, \ldots, \mathbf{v}_r$. But by construction, $\nabla \mathcal{F}(\mathbf{X}^\star)$ has singular values $\boldsymbol{\mu}$ and singular vectors $-\mathbf{u}_1, \ldots, -\mathbf{u}_n$ and $\mathbf{v}_1, \ldots, \mathbf{v}_m$. Thus

$$\nabla \mathcal{F}(\mathbf{X}^\star) + \sum_{i=1}^{\min(n,m)} \boldsymbol{\mu}_i \mathbf{u}_i \mathbf{v}_i^\top = \sum_{i=1}^{r} \boldsymbol{\mu}_i(-\mathbf{u}_i)\mathbf{v}_i^\top + \sum_{i=1}^{r} \boldsymbol{\mu}_i \mathbf{u}_i \mathbf{v}_i^\top = \mathbf{0}.$$

- (primal feasibility) $\|\lambda \mathbf{X}^\star\|_{\mathrm{op}} \leq 1$ by assumption.

- (dual feasibility) $\boldsymbol{\mu} \geq \mathbf{0}$ entrywise by the nonnegativity of singular values.

- (complementary slackness) for $i \in \{1, 2, \ldots, \min(n,m)\}$, if $\boldsymbol{\mu}_i = 0$, then the condition holds. Otherwise, $\boldsymbol{\mu}_i = \boldsymbol{\sigma}_i(\nabla \mathcal{F}(\mathbf{X}^\star)) > 0$ implies $\boldsymbol{\sigma}_i(\lambda \mathbf{X}^\star) = 1$, so the condition holds.

$\square$

## 7.2 Convergence rate of Lion-$\mathcal{K}$ with deterministic gradient

Our analysis in this section is an extension of the discrete-time analysis in Chen et al. (2024), which is in turn inspired by the Lyapunov function for continuous-time LION-$\mathcal{K}$ dynamics (cf. Appendix A).

**Proposition 3.** *Under Assumptions 1, 2, and 5, let $0 \leq \beta_1 < \beta_2 < 1$ and $\lambda > 0$, and suppose $\mathbf{X}_t$, $\mathbf{M}_t$, and $\widetilde{\mathbf{M}}_t$ are updated using* (3). *Let*

$$\begin{aligned} c_t &:= \frac{\eta_t \lambda \beta_1}{\eta_t \lambda (1 - \beta_1) + (1 - \beta_2)} \\ b_t &:= \frac{\beta_1(1 - \beta_2)}{(\beta_2 - \beta_1)(\eta_t \lambda(1 - \beta_1) + (1 - \beta_2))} \\ a_t &:= c_t + 1 \\ \mathcal{H}_t &:= \mathcal{F}(\mathbf{X}_t) - \mathcal{F}^\star + \frac{1}{\lambda}\mathcal{K}^*(\lambda \mathbf{X}_t) + \frac{c_t}{\lambda}(\mathcal{K}^*(\lambda \mathbf{X}_t) + \mathcal{K}(\mathbf{M}_t) - \langle \lambda \mathbf{X}_t, \mathbf{M}_t \rangle) \\ \Gamma_t &:= \left\langle \nabla \mathcal{K}(\widetilde{\mathbf{M}}_{t+1}) - \lambda \mathbf{X}_{t+1}, \widetilde{\mathbf{M}}_{t+1} - \nabla \mathcal{K}^*(\lambda \mathbf{X}_{t+1}) \right\rangle \\ \Delta_t &:= \left\langle \nabla \mathcal{K}(\widetilde{\mathbf{M}}_{t+1}) - \nabla \mathcal{K}(\mathbf{M}_{t+1}), \widetilde{\mathbf{M}}_{t+1} - \mathbf{M}_{t+1} \right\rangle. \end{aligned} \tag{12}$$

*Then for all $T > 0$,*

$$\frac{1}{T}\sum_{t=0}^{T-1} \eta_t(a_t\Gamma_t + b_t\Delta_t) \le \frac{\mathcal{H}_0 - \mathcal{H}_T}{T} + \frac{L}{2T}\sum_{t=0}^{T-1}\eta_t^2 \left\|\nabla\mathcal{K}(\widetilde{\mathbf{M}}_{t+1}) - \lambda\mathbf{X}_{t+1}\right\|_{\mathrm{F}}^2. \tag{13}$$

*Proof.* By smoothness, we have

$$\mathcal{F}(\mathbf{X}_{t+1}) - \mathcal{F}(\mathbf{X}_t) \le \langle\nabla\mathcal{F}(\mathbf{X}_t), \mathbf{X}_{t+1} - \mathbf{X}_t\rangle + \frac{L}{2}\|\mathbf{X}_{t+1} - \mathbf{X}_t\|_{\mathrm{F}}^2. \tag{14}$$

By convexity, we have

$$\begin{aligned}
\mathcal{K}^*(\lambda\mathbf{X}_{t+1}) - \mathcal{K}^*(\lambda\mathbf{X}_t) &\le \langle\lambda\nabla\mathcal{K}^*(\lambda\mathbf{X}_{t+1}), \mathbf{X}_{t+1} - \mathbf{X}_t\rangle \\
\mathcal{K}(\mathbf{M}_{t+1}) - \mathcal{K}(\mathbf{M}_t) &\le \langle\nabla\mathcal{K}(\mathbf{M}_{t+1}), \mathbf{M}_{t+1} - \mathbf{M}_t\rangle.
\end{aligned} \tag{15}$$

Finally, we have

$$\langle\mathbf{X}_{t+1}, \mathbf{M}_{t+1}\rangle - \langle\mathbf{X}_t, \mathbf{M}_t\rangle = \langle\mathbf{M}_t, \mathbf{X}_{t+1} - \mathbf{X}_t\rangle + \langle\mathbf{X}_{t+1}, \mathbf{M}_{t+1} - \mathbf{M}_t\rangle. \tag{16}$$

Combining (14), (15), and (16) gives

$$\mathcal{H}_{t+1} - \mathcal{H}_t \le \langle\nabla_{\mathbf{X}}\mathcal{H}_t, \mathbf{X}_{t+1} - \mathbf{X}_t\rangle + \langle\nabla_{\mathbf{M}}\mathcal{H}_t, \mathbf{M}_{t+1} - \mathbf{M}_t\rangle + \frac{L}{2}\|\mathbf{X}_{t+1} - \mathbf{X}_t\|_{\mathrm{F}}^2, \tag{17}$$

where

$$\nabla_{\mathbf{X}}\mathcal{H}_t := \nabla\mathcal{F}(\mathbf{X}_t) + (1 + c_t)\nabla\mathcal{K}^*(\lambda\mathbf{X}_{t+1}) - c_t\mathbf{M}_t \quad \text{and} \quad \nabla_{\mathbf{M}}\mathcal{H}_t := \frac{c_t}{\lambda}\nabla\mathcal{K}(\mathbf{M}_{t+1}) - c_t\mathbf{X}_{t+1}.$$

Recalling (3), we have

$$\mathbf{X}_{t+1} - \mathbf{X}_t = \eta_t\boldsymbol{\delta}_t \text{ and } \mathbf{M}_{t+1} - \mathbf{M}_t = \frac{1 - \beta_2}{\beta_2 - \beta_1}\left(\widetilde{\mathbf{M}}_{t+1} - \mathbf{M}_{t+1}\right),$$

where $\boldsymbol{\delta}_t := \nabla\mathcal{K}(\widetilde{\mathbf{M}}_{t+1}) - \lambda\mathbf{X}_{t+1}$. Substituting into (17),

$$\begin{aligned}
\mathcal{H}_{t+1} - \mathcal{H}_t &\le \eta_t\langle\nabla_{\mathbf{X}}\mathcal{H}_t, \boldsymbol{\delta}_t\rangle + \frac{1 - \beta_2}{\beta_2 - \beta_1}\left\langle\nabla_{\mathbf{M}}\mathcal{H}_t, \widetilde{\mathbf{M}}_{t+1} - \mathbf{M}_{t+1}\right\rangle + \frac{L}{2}\|\mathbf{X}_{t+1} - \mathbf{X}_t\|_{\mathrm{F}}^2 \\
&= \eta_t\langle\boldsymbol{\delta}_t, a_t\nabla\mathcal{K}^*(\lambda\mathbf{X}_{t+1}) - ((a_t + b_t)\beta_1 - b_t\beta_2)\mathbf{M}_t + (a_t - (a_t + b_t)\beta_1 + b_t\beta_2)\nabla\mathcal{F}(\mathbf{X}_t)\rangle \\
&\quad + \frac{b_t\eta_t\lambda}{c_t}\left\langle\frac{c_t}{\lambda}\nabla\mathcal{K}(\mathbf{M}_{t+1}) - c_t\mathbf{X}_{t+1}, \widetilde{\mathbf{M}}_{t+1} - \mathbf{M}_{t+1}\right\rangle + \frac{L}{2}\|\mathbf{X}_{t+1} - \mathbf{X}_t\|_{\mathrm{F}}^2 \\
&= -\eta_t\left\langle\boldsymbol{\delta}_t, a_t(\widetilde{\mathbf{M}}_{t+1} - \nabla\mathcal{K}^*(\lambda\mathbf{X}_{t+1})) + b_t(\widetilde{\mathbf{M}}_{t+1} - \mathbf{M}_{t+1})\right\rangle \\
&\quad + b_t\eta_t\left\langle\nabla\mathcal{K}(\mathbf{M}_{t+1}) - \lambda\mathbf{X}_{t+1}, \widetilde{\mathbf{M}}_{t+1} - \mathbf{M}_{t+1}\right\rangle + \frac{L}{2}\|\mathbf{X}_{t+1} - \mathbf{X}_t\|_{\mathrm{F}}^2 \\
&= -a_t\eta_t\Gamma_t - b_t\eta_t\left(\left\langle\boldsymbol{\delta}_t, \widetilde{\mathbf{M}}_{t+1} - \mathbf{M}_{t+1}\right\rangle - \left\langle\nabla\mathcal{K}(\mathbf{M}_{t+1}) - \lambda\mathbf{X}_{t+1}, \widetilde{\mathbf{M}}_{t+1} - \mathbf{M}_{t+1}\right\rangle\right) \\
&\quad + \frac{L}{2}\|\mathbf{X}_{t+1} - \mathbf{X}_t\|_{\mathrm{F}}^2 \\
&= -\eta_t(a_t\Gamma_t + b_t\Delta_t) + \frac{\eta_t^2 L}{2}\|\boldsymbol{\delta}_t\|_{\mathrm{F}}^2,
\end{aligned} \tag{18}$$

where the second line uses $c_t = (a_t + b_t)\beta_1 - b_t\beta_2$ and $c_t = a_t - 1$ and the fourth line uses

$$\widetilde{\mathbf{M}}_{t+1} - \mathbf{M}_{t+1} = -(\beta_2 - \beta_1)(\mathbf{M}_t + \nabla\mathcal{F}(\mathbf{X}_t)).$$

Rearranging (18), summing over $T$ iterations, and dividing both sides by $T$ gives the result. $\square$

Proposition 3 establishes bounds for general LION-$\mathcal{K}$ optimizers in the discrete-time setting, and we will use this result to bound the convergence rate of the KKT score function (6).

To avoid tedium in the analysis, we assume that $\mathbf{X}_0$ is initialized so that $\|\lambda\mathbf{X}_0\|_{\mathrm{op}} \leq 1$. Note that by Proposition 1, this implies $\|\lambda\mathbf{X}_t\|_{\mathrm{op}} \leq 1$ for all $t \geq 0$. We remark that in practical implementations of MUON, the weight decay parameter $\lambda$ is known, so $\mathbf{X}_0$ can always be chosen to satisfy $\|\lambda\mathbf{X}_0\|_{\mathrm{op}} \leq 1$.

We state several helper lemmas and defer their proofs to Appendix B.

**Lemma 1.** *Let $\mathcal{K}, \mathcal{K}^* : \mathbb{X} \to \mathbb{R} \cup \{\infty\}$ be a convex, closed, and proper pair of conjugate functions with subgradients $\nabla\mathcal{K}$ and $\nabla\mathcal{K}^*$. Then for all $\mathbf{X}, \mathbf{Y} \in \mathbb{X}$,*

$$\langle \nabla\mathcal{K}(\mathbf{X}) - \nabla\mathcal{K}(\mathbf{Y}), \mathbf{X} - \mathbf{Y} \rangle \geq 0 \tag{19}$$

$$\langle \nabla\mathcal{K}(\mathbf{X}) - \mathbf{Y}, \mathbf{X} - \nabla\mathcal{K}^*(\mathbf{Y}) \rangle \geq 0. \tag{20}$$

**Lemma 2.** *Let $\mathcal{K}(\mathbf{X}) = \|\mathbf{X}\|$ for a norm $\|\cdot\|$ on $\mathbb{X}$. Then $\langle \nabla\mathcal{K}(\mathbf{X}), \mathbf{X} \rangle = \mathcal{K}(\mathbf{X})$.*

**Lemma 3.** *In the setting of Proposition 3, let $\mathcal{K}(\mathbf{X}) = \|\mathbf{X}\|_{\mathrm{tr}}$, $\nabla\mathcal{K}(\mathbf{X}) = \mathrm{msgn}(\mathbf{X})$, $\|\lambda\mathbf{X}_0\|_{\mathrm{op}} \leq 1$, $\eta_t = \eta$, and $C_\mathcal{K} := \sqrt{\min(n,m)}$. Then for all $t > 0$,*

$$\left\|\nabla\mathcal{F}(\mathbf{X}_t) + \widetilde{\mathbf{M}}_t\right\|_{\mathrm{F}} \leq \frac{2\eta C_\mathcal{K} L(1 + \beta_1 - \beta_2)}{1 - \beta_2} + \beta_1\beta_2^{t-1}\|\nabla\mathcal{F}(\mathbf{X}_0) + \mathbf{M}_0\|_{\mathrm{F}}.$$

**Proposition 4.** *In the setting of Lemma 3, for all $T > 0$,*

$$\frac{1}{T}\sum_{t=1}^T \mathcal{S}(\mathbf{X}_t) \leq \frac{\mathcal{H}_0 - \mathcal{H}_T}{\eta T} + 2\eta C_\mathcal{K}^2 L + \frac{2\beta_1 C_\mathcal{K}\|\nabla\mathcal{F}(\mathbf{X}_0) + \mathbf{M}_0\|_{\mathrm{F}}}{(1 - \beta_2)T} + \frac{4\eta C_\mathcal{K}^2 L(1 + \beta_1 - \beta_2)}{1 - \beta_2}. \tag{21}$$

*Proof.* Using the notation of Proposition 3, we have

$$\begin{aligned}
\mathcal{S}(\mathbf{X}_t) &= \langle \mathrm{msgn}(\nabla\mathcal{F}(\mathbf{X}_t)) + \lambda\mathbf{X}_t, \nabla\mathcal{F}(\mathbf{X}_t) \rangle \\
&= \left\langle \mathrm{msgn}(\widetilde{\mathbf{M}}_t) - \lambda\mathbf{X}_t, \widetilde{\mathbf{M}}_t \right\rangle - \left\langle \mathrm{msgn}(\widetilde{\mathbf{M}}_t) - \lambda\mathbf{X}_t, \nabla\mathcal{F}(\mathbf{X}_t) + \widetilde{\mathbf{M}}_t \right\rangle \\
&\quad + \left\langle -\mathrm{msgn}(\nabla\mathcal{F}(\mathbf{X}_t)) - \mathrm{msgn}(\widetilde{\mathbf{M}}_t), \widetilde{\mathbf{M}}_t \right\rangle \\
&\quad - \left\langle -\mathrm{msgn}(\nabla\mathcal{F}(\mathbf{X}_t)) - \mathrm{msgn}(\widetilde{\mathbf{M}}_t), \nabla\mathcal{F}(\mathbf{X}_t) + \widetilde{\mathbf{M}}_t \right\rangle \\
&\leq \left\langle \mathrm{msgn}(\widetilde{\mathbf{M}}_t) - \lambda\mathbf{X}_t, \widetilde{\mathbf{M}}_t \right\rangle - \left\langle \mathrm{msgn}(\widetilde{\mathbf{M}}_t) - \lambda\mathbf{X}_t, \nabla\mathcal{F}(\mathbf{X}_t) + \widetilde{\mathbf{M}}_t \right\rangle \\
&\quad - \left\langle -\mathrm{msgn}(\nabla\mathcal{F}(\mathbf{X}_t)) - \mathrm{msgn}(\widetilde{\mathbf{M}}_t), \nabla\mathcal{F}(\mathbf{X}_t) + \widetilde{\mathbf{M}}_t \right\rangle \\
&\leq \Gamma_{t-1} + \left\langle \mathrm{msgn}(\nabla\mathcal{F}(\mathbf{X}_t)) + \lambda\mathbf{X}_t, \nabla\mathcal{F}(\mathbf{X}_t) + \widetilde{\mathbf{M}}_t \right\rangle \\
&\leq \Gamma_{t-1} + \|\mathrm{msgn}(\nabla\mathcal{F}(\mathbf{X}_t)) + \lambda\mathbf{X}_t\|_{\mathrm{F}} \left\|\nabla\mathcal{F}(\mathbf{X}_t) + \widetilde{\mathbf{M}}_t\right\|_{\mathrm{F}} \\
&\leq \Gamma_{t-1} + 2C_\mathcal{K}\left(\frac{2\eta C_\mathcal{K} L(1 + \beta_1 - \beta_2)}{1 - \beta_2} + \beta_1\beta_2^{t-1}\|\nabla\mathcal{F}(\mathbf{X}_0) + \mathbf{M}_0\|_{\mathrm{F}}\right) \\
&= \Gamma_{t-1} + \frac{4\eta C_\mathcal{K}^2 L(1 + \beta_1 - \beta_2)}{1 - \beta_2} + 2\beta_1\beta_2^{t-1}C_\mathcal{K}\|\nabla\mathcal{F}(\mathbf{X}_0) + \mathbf{M}_0\|_{\mathrm{F}},
\end{aligned}$$

where the first line uses Fact 2 and Lemma 2, the fifth line uses

$$\begin{aligned}
\left\langle \mathrm{msgn}(\nabla\mathcal{F}(\mathbf{X}_t)) + \mathrm{msgn}(\widetilde{\mathbf{M}}_t), \widetilde{\mathbf{M}}_t \right\rangle &= \left\langle \mathrm{msgn}(\nabla\mathcal{F}(\mathbf{X}_t)), \widetilde{\mathbf{M}}_t \right\rangle + \left\|\widetilde{\mathbf{M}}_t\right\|_{\mathrm{tr}} \\
&\geq \left\|\widetilde{\mathbf{M}}_t\right\|_{\mathrm{tr}} - \left\|\widetilde{\mathbf{M}}_t\right\|_{\mathrm{tr}}\|\mathrm{msgn}(\nabla\mathcal{F}(\mathbf{X}_t))\|_{\mathrm{op}} \\
&\geq 0,
\end{aligned}$$

the seventh line uses

$$\partial \mathcal{K}^*(\mathbf{Y}) = \begin{cases} \{\mathbf{0}\} & \text{if } \|\mathbf{Y}\|_{\text{op}} < 1 \\ \{\mathbf{Z} \in \mathbb{X} \mid \langle \mathbf{Z}, \mathbf{Y} \rangle = \|\mathbf{Z}\|_{\text{tr}}\} & \text{if } \|\mathbf{Y}\|_{\text{op}} = 1 \end{cases},$$

the eighth line uses Cauchy–Schwarz, and the ninth line uses Lemma 3. It follows that

$$\begin{aligned}
\frac{1}{T} \sum_{t=1}^{T} \mathcal{S}(\mathbf{X}_t) &\leq \frac{1}{T} \sum_{t=1}^{T} \left( \Gamma_{t-1} + 2\beta_1 \beta_2^{t-1} C_{\mathcal{K}} \|\nabla \mathcal{F}(\mathbf{X}_0) + \mathbf{M}_0\|_{\text{F}} \right) + \frac{4\eta C_{\mathcal{K}}^2 L(1 + \beta_1 - \beta_2)}{1 - \beta_2} \\
&\leq \frac{1}{T} \sum_{t=0}^{T-1} \left( a_t \Gamma_t + b_t \Delta_t + 2\beta_1 \beta_2^t C_{\mathcal{K}} \|\nabla \mathcal{F}(\mathbf{X}_0) + \mathbf{M}_0\|_{\text{F}} \right) + \frac{4\eta C_{\mathcal{K}}^2 L(1 + \beta_1 - \beta_2)}{1 - \beta_2} \\
&\leq \frac{\mathcal{H}_0 - \mathcal{H}_T}{\eta T} + \frac{\eta L}{2T} \sum_{t=0}^{T-1} \left\| \text{msgn}(\widetilde{\mathbf{M}}_{t+1}) - \lambda \mathbf{X}_{t+1} \right\|_{\text{F}}^2 \\
&\quad + \frac{2\beta_1 C_{\mathcal{K}} \|\nabla \mathcal{F}(\mathbf{X}_0) + \mathbf{M}_0\|_{\text{F}}}{(1 - \beta_2)T} + \frac{4\eta C_{\mathcal{K}}^2 L(1 + \beta_1 - \beta_2)}{1 - \beta_2} \\
&\leq \frac{\mathcal{H}_0 - \mathcal{H}_T}{\eta T} + 2\eta C_{\mathcal{K}}^2 L + \frac{2\beta_1 C_{\mathcal{K}} \|\nabla \mathcal{F}(\mathbf{X}_0) + \mathbf{M}_0\|_{\text{F}}}{(1 - \beta_2)T} + \frac{4\eta C_{\mathcal{K}}^2 L(1 + \beta_1 - \beta_2)}{1 - \beta_2},
\end{aligned}$$

where the third line uses Proposition 3 and the fourth line uses

$$\sum_{t=0}^{T-1} \beta_2^t \leq \sum_{j=0}^{\infty} \beta_2^j = \frac{1}{1 - \beta_2}.$$

$\square$

Our convergence rates in the deterministic gradient setting follow directly from the previous results.

**Theorem 3.** *Under Assumptions 1, 2, and 5, let $0 \leq \beta_1 < \beta_2 < 1$, $\lambda > 0$, $\eta_t = \eta = \Theta\left(\frac{1}{\sqrt{T}}\right)$, and $\|\lambda \mathbf{X}_0\|_{\text{op}} \leq 1$, and suppose $\mathbf{X}_t$, $\mathbf{M}_t$, and $\widetilde{\mathbf{M}}_t$ are updated using (3) with deterministic gradients and that $\nabla \mathcal{K}$ has bounded norm. Then*

$$\frac{1}{T} \sum_{t=1}^{T} \left\langle \nabla \mathcal{K}(\widetilde{\mathbf{M}}_t) - \lambda \mathbf{X}_t, \widetilde{\mathbf{M}}_t - \nabla \mathcal{K}^*(\lambda \mathbf{X}_t) \right\rangle = O\left(\frac{1}{\sqrt{T}}\right)$$

$$\frac{1}{T} \sum_{t=1}^{T} \left\langle \nabla \mathcal{K}(\widetilde{\mathbf{M}}_t) - \nabla \mathcal{K}(\mathbf{M}_t), \widetilde{\mathbf{M}}_t - \mathbf{M}_t \right\rangle = O\left(\frac{1}{\sqrt{T}}\right).$$

*Moreover, when $\mathbf{X}_t$, $\mathbf{M}_t$, and $\widetilde{\mathbf{M}}_t$ are updated using (2) with deterministic gradients,*

$$\frac{1}{T} \sum_{t=1}^{T} \mathcal{S}(\mathbf{X}_t) = O\left(\frac{1}{\sqrt{T}}\right).$$

*Proof.* In the setting of Proposition 3, note that $\eta = \Theta\left(\frac{1}{\sqrt{T}}\right)$ and both $\nabla \mathcal{K}(\widetilde{\mathbf{M}})$ and $\lambda \mathbf{X}$ having bounded norm implies that the right-hand side of (13) is $O\left(\frac{1}{T}\right)$. The claim follows after dividing both sides by $\eta$ and realizing $\Gamma_t, \Delta_t \geq 0$ by Lemma 1 and $a_t, b_t \geq 0$.

Now in the setting of Proposition 4, we have that the right-hand side of (21) is $O\left(\frac{1}{\sqrt{T}}\right)$. The claim follows upon realizing

$$\mathcal{S}(\mathbf{X}_t) = \|\nabla \mathcal{F}(\mathbf{X}_t)\|_{\text{tr}} + \langle \lambda \mathbf{X}_t, \nabla \mathcal{F}(\mathbf{X}_t) \rangle \geq \|\nabla \mathcal{F}(\mathbf{X}_t)\|_{\text{tr}} - \|\lambda \mathbf{X}_t\|_{\text{op}} \|\nabla \mathcal{F}(\mathbf{X}_t)\|_{\text{tr}} \geq 0. \tag{22}$$

$\square$

### 7.3 Convergence rate of Lion-$\mathcal{K}$ with stochastic gradient

We now show results analogous to the ones in Section 7.2 when using (3) with stochastic gradients. The following lemmas will be useful for bounding the noise arising from stochastic gradients.

**Lemma 4.** *Let $\mathcal{K}(\mathbf{X}) = \|\mathbf{X}\|$ for a norm $\|\cdot\|$ on $\mathbb{X}$. Then for all $\mathbf{X} \in \mathbb{X}$, $\mathcal{K}^*(\nabla\mathcal{K}(\mathbf{X})) = 0$.*

**Lemma 5.** *Let $\mathbf{X}, \mathbf{Y}$ be $\mathbb{X}$-valued random variables satisfying $\mathrm{Var}(\mathbf{Y}) < \infty$, and let $\mathcal{K}(\mathbf{X}) = \|\mathbf{X}\|$ for a norm $\|\cdot\|$ on $\mathbb{X}$. Then there exists a constant $C_{\mathcal{K}}$ such that*

$$\mathbb{E}\left[\langle \mathbb{E}[\mathbf{Y}] - \mathbf{Y}, \nabla\mathcal{K}(\mathbf{X} + \epsilon\mathbf{Y})\rangle\right] \leq C_{\mathcal{K}}\sqrt{\mathrm{Var}(\mathbf{Y})}.$$

For MUON where $\mathcal{K}(\mathbf{X}) = \|\mathbf{X}\|_{\mathrm{tr}}$ and $\nabla\mathcal{K}(\mathbf{X}) = \mathrm{msgn}(\mathbf{X})$, we can let $C_{\mathcal{K}} = \sqrt{\min(n, m)}$.

**Lemma 6.** *Let $\mathbf{X}, \mathbf{Y}$ be $\mathbb{X}$-valued random variables satisfying*

$$\mathrm{Var}(\mathbf{X}) \leq \sigma^2, \ \mathrm{Var}(\mathbf{Y}) \leq \sigma^2, \ and \ \|\mathbb{E}[\mathbf{X}] - \mathbb{E}[\mathbf{Y}]\|_{\mathrm{F}} \leq R.$$

*Then $\mathbb{E}[\|\mathbf{X} - \mathbf{Y}\|_{\mathrm{F}}] \leq 2\sigma + R$.*

**Lemma 7.** *Let $\mathbf{G}$, $\mathbf{X}$, and $\mathbf{Y}$ be $\mathbb{X}$-valued random variables satisfying $\mathbb{E}[\mathbf{G} \mid \mathbf{X}] = \mathbf{Y}$. Then*

$$\mathbb{E}\left[\|\mathbf{Y}\|_{\mathrm{tr}} + \langle\lambda\mathbf{X}, \mathbf{Y}\rangle\right] \leq \mathbb{E}\left[\|\mathbf{G}\|_{\mathrm{tr}} + \langle\lambda\mathbf{X}, \mathbf{G}\rangle\right].$$

The following result is a stochastic analog of Theorem 3.

**Theorem 4.** *In the setting of Theorem 3 and under Assumption 3, let $\mathcal{K}$ be a norm on $\mathbb{X}$, and suppose instead that $\mathbf{X}_t$, $\mathbf{M}_t$, and $\widetilde{\mathbf{M}}_t$ are updated using (3) with stochastic gradients. Then*

$$\frac{1}{T}\sum_{t=1}^{T}\mathbb{E}\left[\left\langle \nabla\mathcal{K}(\widetilde{\mathbf{M}}_t) - \lambda\mathbf{X}_t, \widetilde{\mathbf{M}}_t - \nabla\mathcal{K}^*(\lambda\mathbf{X}_t)\right\rangle\right] = O\left(\frac{1}{\sqrt{T}} + \frac{\sigma}{\sqrt{n_{\mathrm{batch}}}}\right)$$

$$\frac{1}{T}\sum_{t=1}^{T}\mathbb{E}\left[\left\langle \nabla\mathcal{K}(\widetilde{\mathbf{M}}_t) - \nabla\mathcal{K}(\mathbf{M}_t), \widetilde{\mathbf{M}}_t - \mathbf{M}_t\right\rangle\right] = O\left(\frac{1}{\sqrt{T}} + \frac{\sigma}{\sqrt{n_{\mathrm{batch}}}}\right).$$

*Moreover, when $\mathbf{X}_t$, $\mathbf{M}_t$, and $\widetilde{\mathbf{M}}_t$ are updated using (2) with stochastic gradients,*

$$\frac{1}{T}\sum_{t=1}^{T}\mathbb{E}[\mathcal{S}(\mathbf{X}_t)] = O\left(\frac{1}{\sqrt{T}} + \frac{\sigma}{\sqrt{n_{\mathrm{batch}}}}\right).$$

*Proof.* Let $\boldsymbol{\delta}_t := \nabla\mathcal{K}(\widetilde{\mathbf{M}}_{t+1}) - \lambda\mathbf{X}_{t+1}$. Using the notation of Proposition 3, we have

$$
\begin{aligned}
\mathcal{H}_{t+1} - \mathcal{H}_t &\leq \eta_t\langle\boldsymbol{\delta}_t, a_t\nabla\mathcal{K}^*(\lambda\mathbf{X}_{t+1}) - ((a_t + b_t)\beta_1 - b_t\beta_2)\mathbf{M}_t + (a_t - (a_t + b_t)\beta_1 + b_t\beta_2)\nabla\mathcal{F}(\mathbf{X}_t)\rangle \\
&\quad + b_t\eta_t\left\langle\nabla\mathcal{K}(\mathbf{M}_{t+1}) - \lambda\mathbf{X}_{t+1}, \widetilde{\mathbf{M}}_{t+1} - \mathbf{M}_{t+1}\right\rangle + \frac{L}{2}\|\mathbf{X}_{t+1} - \mathbf{X}_t\|_{\mathrm{F}}^2 \\
&= -\eta_t\left\langle\boldsymbol{\delta}_t, a_t(\widetilde{\mathbf{M}}_{t+1} - \nabla\mathcal{K}^*(\lambda\mathbf{X}_{t+1})) + b_t(\widetilde{\mathbf{M}}_{t+1} - \mathbf{M}_{t+1})\right\rangle + \eta_t\langle\boldsymbol{\delta}_t, \nabla\mathcal{F}(\mathbf{X}_t) - \mathbf{G}_t\rangle \\
&\quad + b_t\eta_t\left\langle\nabla\mathcal{K}(\mathbf{M}_{t+1}) - \lambda\mathbf{X}_{t+1}, \widetilde{\mathbf{M}}_{t+1} - \mathbf{M}_{t+1}\right\rangle + \frac{L}{2}\|\mathbf{X}_{t+1} - \mathbf{X}_t\|_{\mathrm{F}}^2 \\
&= -\eta_t(a_t\Gamma_t + b_t\Delta_t) + \frac{\eta_t^2 L}{2}\|\boldsymbol{\delta}_t\|_{\mathrm{F}}^2 + \eta_t\langle\boldsymbol{\delta}_t, \nabla\mathcal{F}(\mathbf{X}_t) - \mathbf{G}_t\rangle.
\end{aligned}
\tag{23}
$$

It remains to bound $\mathbb{E}[\langle\boldsymbol{\delta}_t, \nabla\mathcal{F}(\mathbf{X}_t) - \mathbf{G}_t\rangle]$. Recalling (3),

$$\boldsymbol{\delta}_t = \frac{1}{1 + \eta_t\lambda}\nabla\mathcal{K}(\widetilde{\mathbf{M}}_{t+1}) - \frac{\lambda}{1 + \eta_t\lambda}\mathbf{X}_t,$$

so

$$
\begin{aligned}
\mathbb{E}[\langle \boldsymbol{\delta}_t, \nabla \mathcal{F}(\mathbf{X}_t) - \mathbf{G}_t \rangle] &= \mathbb{E}\left[\left\langle \frac{1}{1+\eta_t\lambda}\nabla\mathcal{K}(\widetilde{\mathbf{M}}_{t+1}) - \frac{\lambda}{1+\eta_t\lambda}\mathbf{X}_t, \nabla\mathcal{F}(\mathbf{X}_t) - \mathbf{G}_t \right\rangle\right] \\
&= \frac{1}{1+\eta_t\lambda}\mathbb{E}\left[\left\langle \nabla\mathcal{K}(\widetilde{\mathbf{M}}_{t+1}), \nabla\mathcal{F}(\mathbf{X}_t) - \mathbf{G}_t \right\rangle\right] - \frac{\lambda}{1+\eta_t\lambda}\mathbb{E}\left[\langle \mathbf{X}_t, \nabla\mathcal{F}(\mathbf{X}_t) - \mathbf{G}_t \rangle\right] \\
&= \frac{1}{1+\eta_t\lambda}\mathbb{E}\left[\langle \nabla\mathcal{K}(\beta_1\mathbf{M}_t - (1-\beta_1)\mathbf{G}_t), \nabla\mathcal{F}(\mathbf{X}_t) - \mathbf{G}_t \rangle\right] \\
&\leq \frac{C_\mathcal{K}\sigma}{(1+\eta_t\lambda)\sqrt{n_{\text{batch}}}},
\end{aligned}
$$

where the last line uses Lemma 5 with $\mathbf{X} \leftarrow \beta_1\mathbf{M}_t$, $\mathbf{Y} \leftarrow \mathbf{G}_t$, $\epsilon \leftarrow -(1-\beta_1)$, and some constant $C_\mathcal{K}$. Substituting into (23),

$$
\mathbb{E}[\mathcal{H}_{t+1} - \mathcal{H}_t] \leq \mathbb{E}\left[-\eta_t(a_t\Gamma_t + b_t\Delta_t) + \frac{\eta_t^2 L}{2}\|\boldsymbol{\delta}_t\|_{\text{F}}^2 + \frac{\eta_t C_\mathcal{K}\sigma}{(1+\eta_t\lambda)\sqrt{n_{\text{batch}}}}\right]. \tag{24}
$$

Taking $\eta_t = \eta$, rearranging (24), summing over $T$ iterations, and dividing both sides by $\eta T$ yields

$$
\frac{1}{T}\sum_{t=0}^{T-1}\mathbb{E}[a_t\Gamma_t + b_t\Delta_t] \leq \mathbb{E}\left[\frac{\mathcal{H}_0 - \mathcal{H}_T}{\eta T} + \frac{\eta L}{2T}\sum_{t=0}^{T-1}\left\|\nabla\mathcal{K}(\widetilde{\mathbf{M}}_{t+1}) - \lambda\mathbf{X}_{t+1}\right\|_{\text{F}}^2 + \frac{C_\mathcal{K}\sigma}{(1+\eta\lambda)\sqrt{n_{\text{batch}}}}\right]. \tag{25}
$$

To show the result for MUON, we adapt the proof of Proposition 4. We have

$$
\begin{aligned}
\mathbb{E}[\|\mathbf{G}_t\|_{\text{tr}} &+ \langle \lambda\mathbf{X}_t, \mathbf{G}_t \rangle] \\
&\leq \mathbb{E}[\Gamma_{t-1}] + \mathbb{E}\left[\|\text{msgn}(\mathbf{G}_t) + \lambda\mathbf{X}_t\|_{\text{F}}\left\|\mathbf{G}_t + \widetilde{\mathbf{M}}_t\right\|_{\text{F}}\right] \\
&\leq \mathbb{E}[\Gamma_{t-1}] + 2C_\mathcal{K}(\mathbb{E}[\|\mathbf{G}_t - \mathbf{G}_{t-1}\|_{\text{F}}] + \beta_1\mathbb{E}[\|\mathbf{G}_{t-1} + \mathbf{M}_{t-1}\|_{\text{F}}]) \\
&\leq \mathbb{E}[\Gamma_{t-1}] + 2C_\mathcal{K}\left(\frac{2\sigma}{\sqrt{n_{\text{batch}}}} + 2\eta C_\mathcal{K}L\right) \\
&\quad + 2\beta_1 C_\mathcal{K}\sum_{k=1}^{t-1}\beta_2^{t-k-1}\mathbb{E}[\|\mathbf{G}_k - \mathbf{G}_{k-1}\|_{\text{F}}] + 2\beta_1\beta_2^{t-1}C_\mathcal{K}\mathbb{E}[\|\mathbf{G}_0 + \mathbf{M}_0\|_{\text{F}}] \\
&\leq \mathbb{E}[\Gamma_{t-1}] + 2C_\mathcal{K}\left(\frac{2\sigma}{\sqrt{n_{\text{batch}}}} + 2\eta C_\mathcal{K}L\right) \\
&\quad + \frac{2\beta_1 C_\mathcal{K}}{1-\beta_2}\left(\frac{2\sigma}{\sqrt{n_{\text{batch}}}} + 2\eta C_\mathcal{K}L\right) + 2\beta_1\beta_2^{t-1}C_\mathcal{K}\mathbb{E}[\|\mathbf{G}_0 + \mathbf{M}_0\|_{\text{F}}] \\
&= \mathbb{E}[\Gamma_{t-1}] + \frac{4C_\mathcal{K}(1+\beta_1-\beta_2)}{1-\beta_2}\left(\frac{\sigma}{\sqrt{n_{\text{batch}}}} + \eta C_\mathcal{K}L\right) + 2\beta_1\beta_2^{t-1}C_\mathcal{K}\mathbb{E}[\|\mathbf{G}_0 + \mathbf{M}_0\|_{\text{F}}],
\end{aligned}
$$

where the fourth and seventh lines use Lemma 6. Now, as before,

$$
\begin{aligned}
\frac{1}{T}\sum_{t=1}^{T}\mathbb{E}[\mathcal{S}(\mathbf{X}_t)] &= \frac{1}{T}\sum_{t=1}^{T}\mathbb{E}[\|\nabla\mathcal{F}(\mathbf{X}_t)\|_{\mathrm{tr}} + \langle\lambda\mathbf{X}_t, \nabla\mathcal{F}(\mathbf{X}_t)\rangle] \le \frac{1}{T}\sum_{t=1}^{T}\mathbb{E}[\|\mathbf{G}_t\|_{\mathrm{tr}} + \langle\lambda\mathbf{X}_t, \mathbf{G}_t\rangle]\\
&\le \frac{1}{T}\sum_{t=1}^{T}\left(\mathbb{E}[\Gamma_{t-1}] + 2\beta_1\beta_2^{t-1}C_{\mathcal{K}}\mathbb{E}[\|\mathbf{G}_0 + \mathbf{M}_0\|_{\mathrm{F}}]\right) + \frac{4C_{\mathcal{K}}(1+\beta_1-\beta_2)}{1-\beta_2}\left(\frac{\sigma}{\sqrt{n_{\mathrm{batch}}}} + \eta C_{\mathcal{K}}L\right)\\
&\le \frac{1}{T}\sum_{t=0}^{T-1}\left(\mathbb{E}[a_t\Gamma_t + b_t\Delta_t] + 2\beta_1\beta_2^t C_{\mathcal{K}}\mathbb{E}[\|\mathbf{G}_0 + \mathbf{M}_0\|_{\mathrm{F}}]\right)\\
&\quad + \frac{4C_{\mathcal{K}}(1+\beta_1-\beta_2)}{1-\beta_2}\left(\frac{\sigma}{\sqrt{n_{\mathrm{batch}}}} + \eta C_{\mathcal{K}}L\right)\\
&\le \mathbb{E}\left[\frac{\mathcal{H}_0 - \mathcal{H}_T}{\eta T}\right] + 2\eta C_{\mathcal{K}}^2 L + \frac{C_{\mathcal{K}}\sigma}{(1+\eta\lambda)\sqrt{n_{\mathrm{batch}}}} + \frac{2\beta_1 C_{\mathcal{K}}\mathbb{E}[\|\mathbf{G}_0 + \mathbf{M}_0\|_{\mathrm{F}}]}{(1-\beta_2)T}\\
&\quad + \frac{4C_{\mathcal{K}}(1+\beta_1-\beta_2)}{1-\beta_2}\left(\frac{\sigma}{\sqrt{n_{\mathrm{batch}}}} + \eta C_{\mathcal{K}}L\right),
\end{aligned}
$$

where the first line uses Lemma 7 and the fifth line uses (25). $\qquad\square$

## 7.4 Convergence of Muon to the set of KKT points

In this section, we use LaSalle's invariance principle (LaSalle, 1960), Proposition 2, and Theorem 4 to show that MUON converges to the set of KKT points of (5).

**Definition 4** ($\omega$-limit set). *Let $\{\mathbf{X}_t\}_{t\in\mathbb{N}}$ be a stochastic process. A set $\mathbb{M}$ is called the $\omega$-limit set of the process if it is the minimal closed set satisfying*

$$
\Pr\left(\lim_{t\to\infty}d(\mathbf{X}_t, \mathbb{M}) = 0\right) = 1.
$$

In other words, $\mathbb{M}$ is the smallest closed set such that the trajectories approach $\mathbb{M}$ a.s. as $t\to\infty$. This is equivalent to $\mathbb{M}$ being the support of the union of all limit measures of $\{\mathbf{X}_t\}_{t\in\mathbb{N}}$.

**Lemma 8.** *Let $X$ be a nonnegative random variable such that $\mathbb{E}[X] = 0$. Then $X = 0$ a.s.*

**Lemma 9** (LaSalle's invariance principle for stochastic dynamical systems). *Let $\{\mathbf{X}_t\}_{t\in\mathbb{N}}$ be a stochastic process contained in a bounded set a.s., and suppose there exist a nonnegative function $\mathcal{V}$ and a nonnegative, lower semicontinuous function $h$ such that*

$$
\mathbb{E}[\mathcal{V}(\mathbf{X}_{t+1}) \mid \mathcal{F}_t] - \mathcal{V}(\mathbf{X}_t) \le -\alpha_t h(\mathbf{X}_{t+\ell}) + \gamma_t \ a.s.,
$$

*where $\{\mathcal{F}_t\}_{t\in\mathbb{N}}$ is the natural filtration of $\{\mathbf{X}_t\}_{t\in\mathbb{N}}$, $\ell\in\mathbb{N}$, and the nonnegative sequences $\{\alpha_t\}_{t\in\mathbb{N}}$ and $\{\gamma_t\}_{t\in\mathbb{N}}$ satisfy*

$$
\sum_{t=0}^{\infty}\alpha_t = \infty \quad and \quad \sum_{t=0}^{\infty}\gamma_t < \infty.
$$

*Let $\mathbb{M}$ be the $\omega$-limit set of $\{\mathbf{X}_t\}_{t\in\mathbb{N}}$. Then $\mathbb{M}$ is contained in the set $\{\mathbf{X}\in\mathbb{X} \mid h(\mathbf{X}) = 0\}$.*

**Theorem 5.** *Under Assumptions 1, 2, and 4, let $0 \le \beta_1 < \beta_2 < 1$, $\lambda > 0$, $\eta_t \le \frac{1}{\lambda}$, and $\sum_{t=0}^{\infty}\eta_t = \infty$, and suppose $\mathbf{X}_t$, $\mathbf{M}_t$, and $\widetilde{\mathbf{M}}_t$ are updated using (2). Then $\mathbf{X}_t$ converges to*

$$
\mathbb{B} := \{\mathbf{X}\in\mathbb{X} \mid \|\lambda\mathbf{X}\|_{\mathrm{op}} \le 1\} \ a.s.
$$

*Proof.* If $\mathbf{X}_t$ enters $\mathbb{B}$ within a finite amount of time, then it remains there thenceforth by Proposition 1. Now suppose $\mathbf{X}_t$ does not enter $\mathbb{B}$ within any finite amount of time. Let

$$
\mathcal{V}(\mathbf{X}) = \max\left(\|\mathbf{X}\|_{\mathrm{op}} - \frac{1}{\lambda}, 0\right), \quad h(\mathbf{X}) = \max\left(\|\mathbf{X}\|_{\mathrm{op}} - \frac{1}{\lambda}, 0\right), \quad \alpha_t = \eta_t\lambda, \quad \text{and} \quad \gamma_t = 0.
$$

We verify that $\mathbf{X}_t$ is bounded a.s., $\mathcal{V}$ is nonnegative, $h$ is nonnegative and lower semicontinuous, $\sum_{t=0}^{\infty} \alpha_t = \infty$, $\sum_{t=0}^{\infty} \gamma_t < \infty$, and

$$\mathbb{E}[\mathcal{V}(\mathbf{X}_{t+1}) \mid \mathbf{X}_t] - \mathcal{V}(\mathbf{X}_t) = \mathbb{E}\left[\|\mathbf{X}_{t+1}\|_{\mathrm{op}} - \frac{1}{\lambda} \;\middle|\; \mathbf{X}_t\right] - \left(\|\mathbf{X}_t\|_{\mathrm{op}} - \frac{1}{\lambda}\right) \leq -\alpha_t h(\mathbf{X}_t) + \gamma_t \text{ a.s.}$$

by Proposition 1. Thus $\mathbf{X}_t$ converges to $\{\mathbf{X} \in \mathbb{X} \mid h(\mathbf{X}) = 0\} = \mathbb{B}$ a.s. by Lemma 9. $\qquad\square$

**Theorem 6.** *In the setting of Theorem 5, let $\|\lambda\mathbf{X}_0\|_{\mathrm{op}} \leq 1$, $\sum_{t=0}^{\infty} \eta_t^2 < \infty$, and*

$$\sum_{t=0}^{\infty} \eta_t \left[\frac{\sigma^2}{n_{\mathrm{batch}}}\right]_t^{\frac{1}{2}} < \infty.$$

*Then $\mathbf{X}_t$ converges to the set of KKT points of (5) a.s.*

*Proof.* Let

$$\mathcal{H}(\mathbf{X}, \mathbf{M}, \widetilde{\mathbf{M}}, \eta) = \mathcal{F}(\mathbf{X}) - \mathcal{F}^{\star} + \frac{\eta\beta_1}{\eta\lambda(1-\beta_1) + (1-\beta_2)}(\|\mathbf{M}\|_{\mathrm{tr}} - \langle\lambda\mathbf{X}, \mathbf{M}\rangle),$$

$$h(\mathbf{X}, \mathbf{M}, \widetilde{\mathbf{M}}, \eta) = \left\|\widetilde{\mathbf{M}}\right\|_{\mathrm{tr}} - \left\langle\lambda\mathbf{X}, \widetilde{\mathbf{M}}\right\rangle, \quad \alpha_t = \eta_t, \quad \text{and} \quad \gamma_t = 2\eta_t^2 C_{\mathcal{K}}^2 L + \eta_t C_{\mathcal{K}}\left[\frac{\sigma^2}{n_{\mathrm{batch}}}\right]_t^{\frac{1}{2}},$$

where $C_{\mathcal{K}} := \sqrt{\min(n, m)}$. Letting $\mathbf{Z}_t := (\mathbf{X}_t, \mathbf{M}_t, \widetilde{\mathbf{M}}_t, \eta_t)$ and using the notation of Proposition 3 with $\mathcal{K}(\mathbf{X}) = \|\mathbf{X}\|_{\mathrm{tr}}$ and $\nabla\mathcal{K}(\mathbf{X}) = \mathrm{msgn}(\mathbf{X})$, we verify that $\mathbf{Z}_t$ is bounded a.s., $\mathcal{H}$ is nonnegative, $h$ is nonnegative and lower semicontinuous, $\sum_{t=0}^{\infty} \alpha_t = \infty$, $\sum_{t=0}^{\infty} \gamma_t < \infty$, and

$$\mathbb{E}[\mathcal{H}(\mathbf{Z}_{t+1}) \mid \mathbf{Z}_t] - \mathcal{H}(\mathbf{Z}_t) \leq -\alpha_t(a_t\Gamma_t + b_t\Delta_t) + \gamma_t \leq -\alpha_t h(\mathbf{Z}_{t+1}) + \gamma_t \text{ a.s.}$$

by (24) and $a_t\Gamma_t + b_t\Delta_t \geq \Gamma_t = h(\mathbf{Z}_{t+1})$. Thus by Lemma 9, $h(\mathbf{Z}_t) = 0$ a.s. as $t \to \infty$, i.e.

$$\lim_{t\to\infty} \left\|\widetilde{\mathbf{M}}_t\right\|_{\mathrm{tr}} - \left\langle\lambda\mathbf{X}_t, \widetilde{\mathbf{M}}_t\right\rangle = 0 \text{ a.s.}$$

It follows that, with probability 1,

$$\begin{aligned}
\lim_{t\to\infty} \mathcal{S}(\mathbf{X}_t) &= \lim_{t\to\infty} \left(\left\|\nabla\mathcal{F}(\mathbf{X}_t) + \widetilde{\mathbf{M}}_t - \widetilde{\mathbf{M}}_t\right\|_{\mathrm{tr}} + \left\langle\lambda\mathbf{X}_t, \nabla\mathcal{F}(\mathbf{X}_t) + \widetilde{\mathbf{M}}_t - \widetilde{\mathbf{M}}_t\right\rangle\right) \\
&\leq \lim_{t\to\infty} \left(\left\|\nabla\mathcal{F}(\mathbf{X}_t) + \widetilde{\mathbf{M}}_t\right\|_{\mathrm{tr}} + \left\|\widetilde{\mathbf{M}}_t\right\|_{\mathrm{tr}} + \left\langle\lambda\mathbf{X}_t, \nabla\mathcal{F}(\mathbf{X}_t) + \widetilde{\mathbf{M}}_t\right\rangle - \left\langle\lambda\mathbf{X}_t, \widetilde{\mathbf{M}}_t\right\rangle\right) \\
&\leq \lim_{t\to\infty} \left(\left\|\nabla\mathcal{F}(\mathbf{X}_t) + \widetilde{\mathbf{M}}_t\right\|_{\mathrm{tr}} + \|\lambda\mathbf{X}_t\|_{\mathrm{op}}\left\|\nabla\mathcal{F}(\mathbf{X}_t) + \widetilde{\mathbf{M}}_t\right\|_{\mathrm{tr}}\right) \leq \lim_{t\to\infty} 2\left\|\nabla\mathcal{F}(\mathbf{X}_t) + \widetilde{\mathbf{M}}_t\right\|_{\mathrm{tr}} \\
&\leq \lim_{t\to\infty} \left(4\eta_{t-1}C_{\mathcal{K}}^2 L + 4\beta_1 C_{\mathcal{K}}^2 L \sum_{k=1}^{t-1}\beta_2^{t-k-1}\eta_{k-1} + 2\beta_1\beta_2^{t-1}C_{\mathcal{K}}\|\nabla\mathcal{F}(\mathbf{X}_0) + \mathbf{M}_0\|_{\mathrm{F}}\right. \\
&\qquad\left. + 2(1-\beta_1)\|\mathbf{G}_{t-1} - \nabla\mathcal{F}(\mathbf{X}_{t-1})\|_{\mathrm{tr}} + 2\beta_1(1-\beta_2)\sum_{k=0}^{t-2}\beta_2^{t-k-2}\|\mathbf{G}_k - \nabla\mathcal{F}(\mathbf{X}_k)\|_{\mathrm{tr}}\right) \\
&= 0,
\end{aligned}$$

where the fourth line uses (31), the fifth line vanishes because of the asymptotically deterministic gradient, and the sixth line uses the Silverman–Toeplitz theorem to show that the sums vanish. By Proposition 2, we conclude that $\mathbf{X}_t$ converges to the set of KKT points of (5) a.s. $\qquad\square$

**Remark 1.** *The KKT conditions for (5) are necessary for optimality, and under the additional assumption that $\mathcal{F}$ is convex, the KKT conditions also become sufficient for optimality. In this case, Theorem 6 shows that the algorithm converges to the set of optimal solutions of (5).*

**Remark 2.** *The $\sum_{t=0}^{\infty} \eta_t \left[ \frac{\sigma^2}{n_{\text{batch}}} \right]_t^{\frac{1}{2}} < \infty$ condition in Theorem 6 can be achieved by using deterministic gradients or sufficiently increasing batch sizes, e.g. $[n_{\text{batch}}]_t = \Theta(t)$ when $\eta_t = O(t^{-1})$.*

**Remark 3.** *Theorem 6 does not guarantee convergence to a single point; compare with SIGNSGD and Frank–Wolfe, which can fail to converge even in the smooth convex setting (Karimireddy et al., 2019; Bolte et al., 2024).*

## 8 Experiments

In this section, we present several empirical studies to validate our theoretical results. Our experiments focus on verifying the implicit constraints enforced by MUON, demonstrating its convergent behavior, and comparing its performance to standard adaptive optimizers such as ADAMW across various tasks, architectures, and choices of $\mathcal{K}$.

### 8.1 Toy example

To clearly illustrate the behavior of MUON, we consider a simplified two-dimensional matrix optimization problem. Figure 2 shows trajectories of singular values under two distinct constraint strengths ($\lambda = 1.5$ and $\lambda = 5.0$) with $\beta_1 = \beta_2 = 0.95$ and $\eta = 0.001$. We initialize $\mathbf{X}$ as a random matrix with singular values $(1, 0)$. The singular values quickly move into and remain within their respective constraint regions, clearly demonstrating the enforcement of spectral norm constraints.

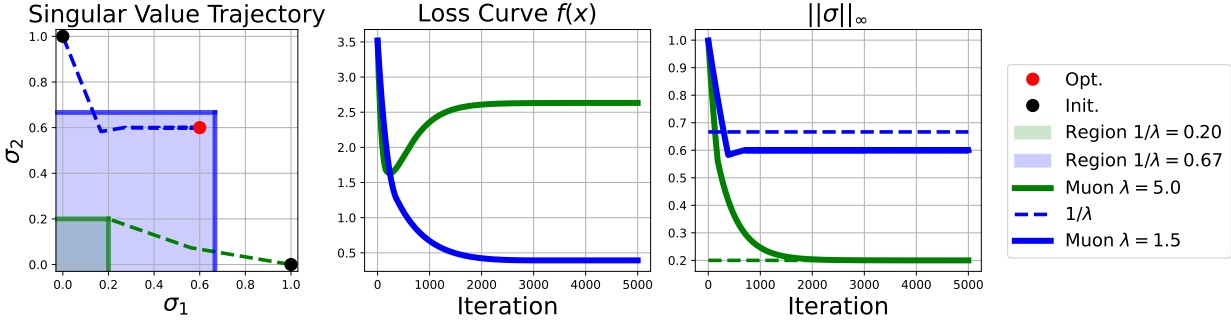

Figure 2: Trajectories of MUON for the matrix optimization problem $\min_{\mathbf{X} \in \mathbb{R}^{2 \times 2}} f(\mathbf{X})$, where $f(\mathbf{X}) = \|\mathbf{AX} - \mathbf{B}\|_F^2 + \mu \|\mathbf{X}\|_F^2$, $\mathbf{A}, \mathbf{B} \in \mathbb{R}^{2 \times 2}$, and $\mu \in \mathbb{R}_{>0}$, evaluated for two different values of $\lambda$: 1.5 (blue) and 5.0 (green). The colored boxes illustrate the constraint sets induced by $\|\mathbf{X}\|_\infty \leq \frac{1}{\lambda}$: the blue box corresponds to $\lambda = 1.5$, and the green box corresponds to $\lambda = 5.0$. The red dot indicates the optimal solution.

### 8.2 Constraint verification

We examine the constraint enforcement of MUON using a toy example (Figure 3). In the upper panel, we set $\lambda = 1.25$ and initialize two trajectories within the feasible region (shaded green): one trajectory starts at $\mathbf{X}^0 = \mathbf{diag}\,(0.01, 0.75)$ (red) and the other at $\mathbf{X}^1 = \mathbf{diag}\,(0.7, 0.7)$ (blue). Both trajectories converge to the feasible optimum, although the objective function exhibits a nonmonotonic spike near iteration 2000. Despite this, the constructed Lyapunov function $\mathcal{H}$ from Proposition 3 decreases monotonically, verifying the theoretical convergence guarantees. In the lower panel, we increase $\lambda$ to 4 and initialize trajectories outside the feasible region at $\mathbf{X}^0 = \mathbf{diag}\,(0.1, 0.95)$ (red curve) and $\mathbf{X}^1 = \mathbf{diag}\,(0.7, 0.9)$ (blue curve). Here, since the optimal solution is infeasible, the trajectories converge to a feasible projection. Although the objective function plateaus around iteration 500, the Lyapunov function (7) continues to decrease monotonically.

We further validate the implicit constraint enforcement of MUON on standard benchmarks. Figure 4 demonstrates the rapid enforcement of singular value constraints in ResNet-18 trained on CIFAR-10. Singular

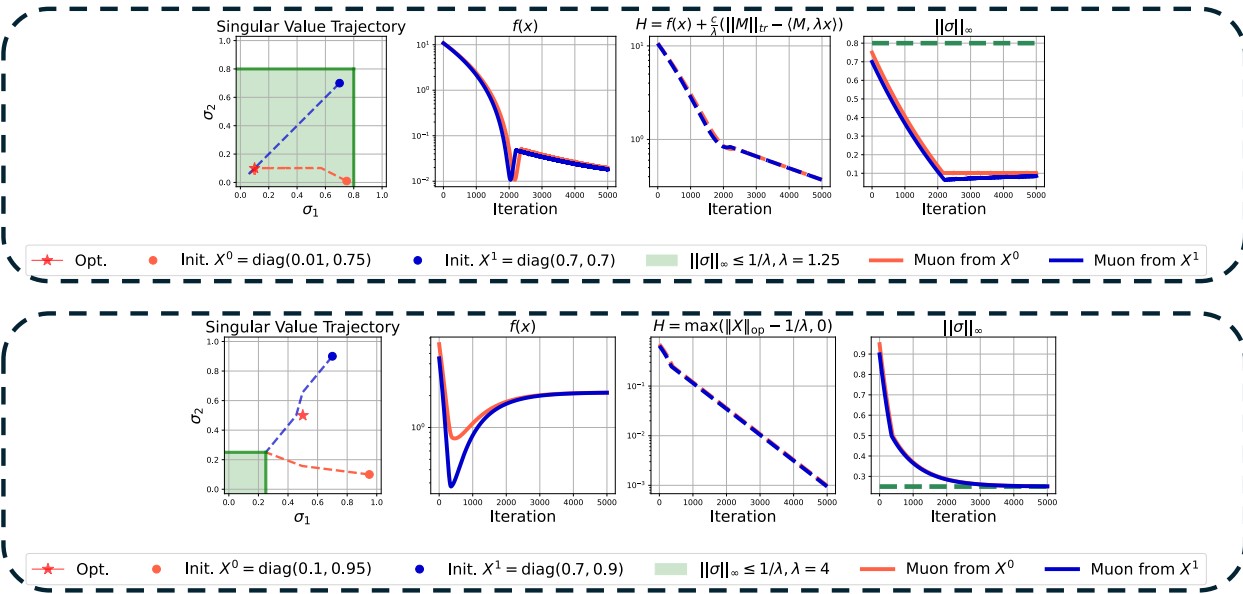

Figure 3: MUON with $\lambda = 1.25$, initialized within the feasible region (upper panel), and $\lambda = 4$, initialized outside the feasible region (lower panel). The green region denotes the constraint region. Both cases illustrate convergence and the monotonic decrease of the Lyapunov function $\mathcal{H}$.

values initially outside the constraint set quickly enter within approximately 400 training steps and remain reliably bounded thereafter.

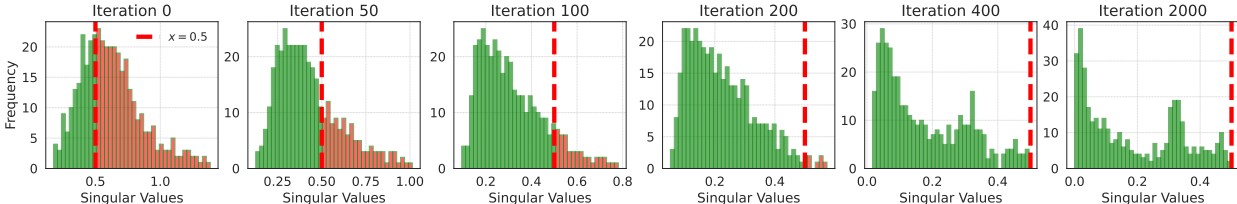

Figure 4: Histograms of singular values of the weight matrices from each module of ResNet-18 trained on CIFAR-10 with the MUON optimizer ($\lambda = 2.0$). The constraint $\|\mathbf{W}\|_{\mathrm{op}} \leq \frac{1}{\lambda}$ (indicated by red vertical lines) is rapidly enforced, with singular values initially outside the constraint region quickly moving inside within approximately 400 training steps. Once inside, singular values remain consistently bounded by the constraint throughout the remainder of training.

In Figure 5, we extend this verification to larger-scale tasks and architectures, including ImageNet classification and language modeling using ResNet-50, ViT-B/16, Qwen-100M, and LLaMA-300M models. The results consistently confirm that the singular values remain bounded within the theoretical upper limit ($\frac{1}{\lambda}$), indicated by horizontal dashed lines, under different regularization strengths ($\lambda = 2.0$ and $\lambda = 4.0$).

## 8.3 Implicit spectral regularization in large models

We investigate the implicit spectral regularization induced by the MUON optimizer in comparison to ADAMW. Figure 6 displays the singular value distributions of converged weights for the query ($\mathbf{W}_Q$), key ($\mathbf{W}_K$), and value ($\mathbf{W}_V$) matrices in the LLaMA 0.5B model trained by both optimizers. We observe that MUON consistently produces regularized singular value distributions, reflecting the effect of its implicit spectral norm constraint. In contrast, singular values from ADAMW do not exhibit any spectral regularization.

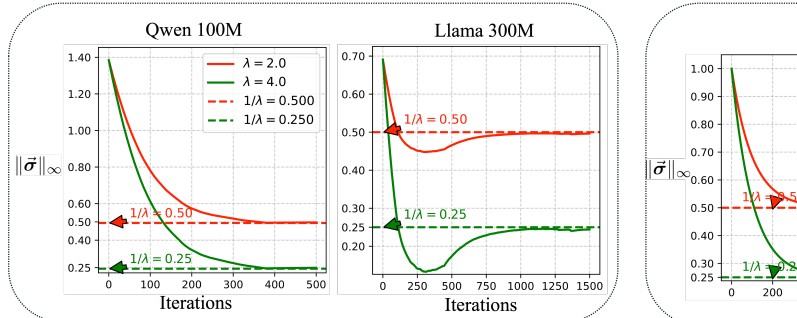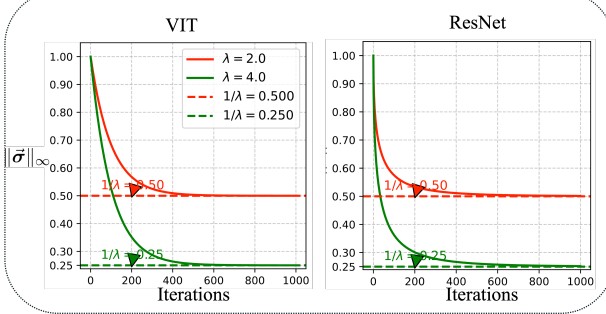

Figure 5: Verification of the implicit constraint enforced by the MUON optimizer with decoupled weight decay on ImageNet and language modeling tasks, across architectures including ResNet-50, ViT-B/16, Qwen-100M, and LLaMA-300M. The red and green curves correspond to the choices $\lambda = 2.0$ and $\lambda = 4.0$, respectively. The horizontal dashed lines indicate the theoretical upper bounds $\frac{1}{\lambda}$ of the implicit box constraints.

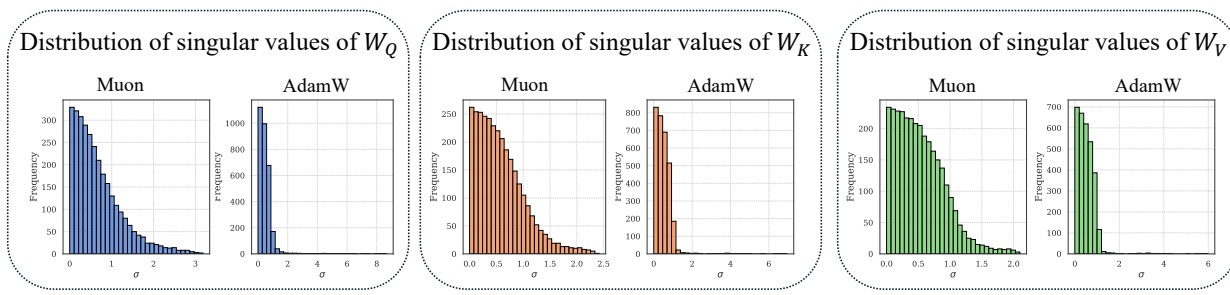

Figure 6: Singular value distributions of converged weights for the query matrix ($\mathbf{W}_Q$), key matrix ($\mathbf{W}_K$), and value matrix ($\mathbf{W}_V$) matrices trained with MUON and ADAMW optimizers on the LLaMA 0.5B model. From left to right, each subfigure compares singular value distributions obtained by MUON and ADAMW, respectively.

### 8.4 Generalizations via different convex functions

We explore the flexibility provided by the LION-$\mathcal{K}$ framework by varying the convex map $\mathcal{K}$. Figure 7 shows results from applying LION-$\mathcal{K}$ with alternative convex maps on the previously defined matrix optimization problem. For instance, selecting the thresholding function

$$\mathcal{K}(\mathbf{X}) = \sum_{i=1}^{\min(n,m)} \max(|\boldsymbol{\sigma}_i(\mathbf{X})| - e, 0)$$

induces a soft-thresholding penalty on singular values exceeding a threshold $e$, illustrating how different convex maps yield distinct implicit constraints and penalty structures.

## 9 Conclusion

In this paper, we showed that MUON is an instance of the LION-$\mathcal{K}$ optimizer when equipped with the nuclear norm and extended the LION-$\mathcal{K}$ analysis of Chen et al. (2024) to accommodate matrix-valued updates in both deterministic and stochastic gradient settings. We tailored our analysis to MUON with decoupled weight decay, demonstrating that it converges to the set of KKT points of a spectral-norm-constrained optimization problem. Beyond theoretical guarantees, we developed a framework for the generalization of MUON and empirically validated our results. Overall, we present MUON as a theoretically grounded optimizer for deep learning, with promising directions for future work.

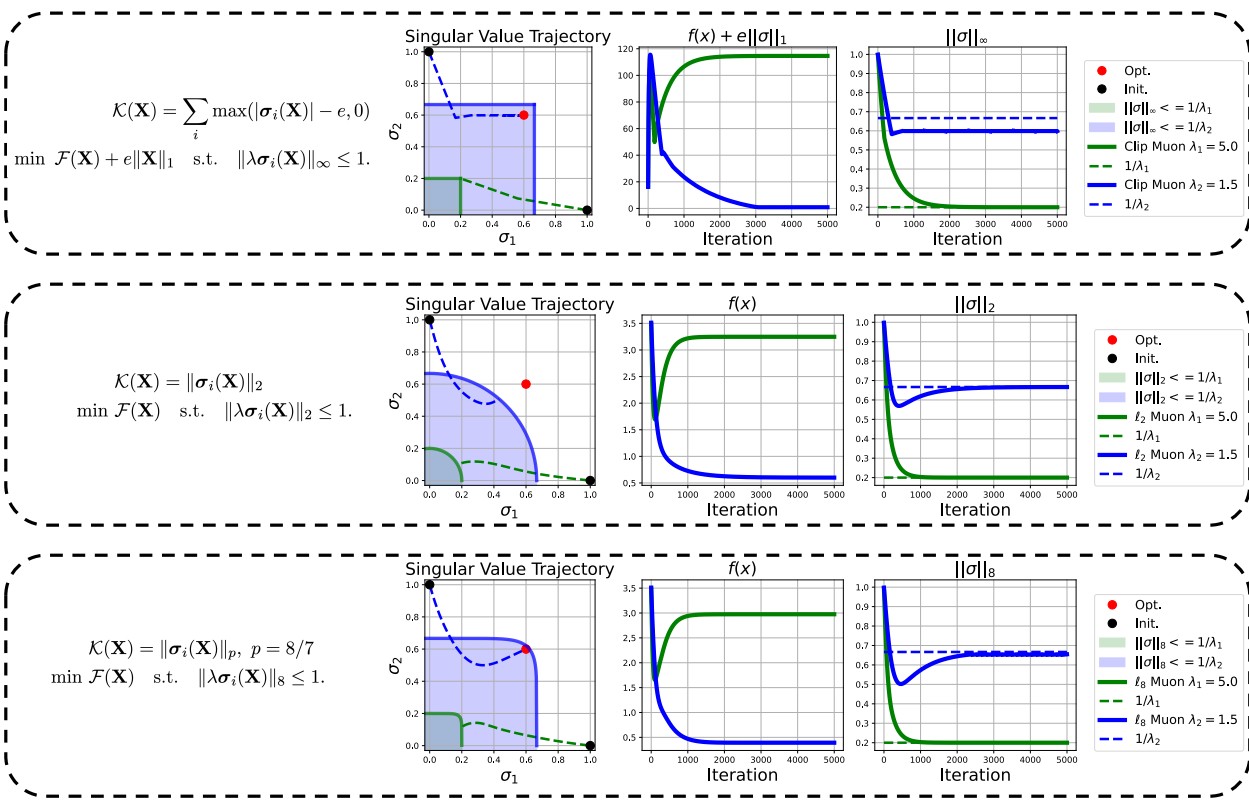

Figure 7: Behavior of the LION-$\mathcal{K}$ optimizer under different choices of $\mathcal{K}$. Using the same objective function as in Figure 2, varying the choice of $\mathcal{K}$ induces distinct implicit constraints and penalty structures on the objective function. The colored boxes illustrate the constraint sets induced by $\mathcal{K}$. For example, choosing the thresholding function $\mathcal{K}(\mathbf{X}) = \sum_{i=1}^{\min(n,m)} \max(|\boldsymbol{\sigma}_i(\mathbf{X})| - e, 0)$ enforces a soft-thresholding penalty on singular values $\boldsymbol{\sigma}_i(\mathbf{X})$ exceeding a threshold $e$. Here, $\|\boldsymbol{\sigma}_i(\mathbf{X})\|_p$ denotes the Schatten $p$-norm of $\mathbf{X}$.

**Limitations.** While our theoretical and empirical findings provide substantial insights, several limitations suggest directions for future research. First, although our analysis focuses on key convex maps, such as the nuclear norm and clipping functions, extending the LION-$\mathcal{K}$ framework to a broader class of convex maps may reveal additional implicit regularization behaviors tailored to specific tasks. Second, extending our results to practical training conditions (e.g. general learning rates, nonsmooth objectives) warrants further investigation. Finally, scaling empirical evaluations to larger models and more diverse tasks would help further validate and refine the practical applicability and robustness of the MUON optimizer and its LION-$\mathcal{K}$ generalizations.

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

# A    Continuous-time analysis of Lion-$\mathcal{K}$

For completeness, we give a straightforward extension of the analysis of Chen et al. (2024) to establish the convergence of the LION-$\mathcal{K}$ ODE (9) for matrices.

**Theorem 7.** *Under Assumptions 1 and 5, let $\mathcal{F}$, $\mathcal{K}$, and $\mathcal{K}^*$ be continuously differentiable and*

$$\dot{\mathbf{X}}_t = \nabla\mathcal{K}(\mathbf{M}_t - \epsilon(\gamma\mathbf{M}_t + \alpha\nabla\mathcal{F}(\mathbf{X}_t))) - \lambda\mathbf{X}_t$$
$$\dot{\mathbf{M}}_t = -\alpha\nabla\mathcal{F}(\mathbf{X}_t) - \gamma\mathbf{M}_t, \tag{26}$$

*where $\alpha, \gamma, \epsilon, \lambda > 0$ and $\epsilon\gamma \leq 1$. Define*

$$\mathcal{H}(\mathbf{X}, \mathbf{M}) := \alpha(\mathcal{F}(\mathbf{X}) - \mathcal{F}^\star) + \frac{\gamma}{\lambda}(\mathcal{K}^*(\lambda\mathbf{X}) + \mathcal{K}(\mathbf{0})) + \frac{1 - \epsilon\gamma}{1 + \epsilon\lambda}(\mathcal{K}^*(\lambda\mathbf{X}) + \mathcal{K}(\mathbf{M}) - \langle\mathbf{M}, \lambda\mathbf{X}\rangle). \tag{27}$$

*Then for all $t$, $\mathcal{H}(\mathbf{X}_t, \mathbf{M}_t) \geq 0$ and $\frac{\mathrm{d}}{\mathrm{d}t}\mathcal{H}(\mathbf{X}_t, \mathbf{M}_t) \leq 0$, i.e. $\mathcal{H}$ is a Lyapunov function for (26).*

*Proof.* For simplicity, we drop the index $t$. By assumption, $\mathcal{F}(\mathbf{X}) - \mathcal{F}^\star \geq 0$, by the Fenchel–Young inequality, $\mathcal{K}^*(\lambda\mathbf{X}) + \mathcal{K}(\mathbf{M}) - \langle\mathbf{M}, \lambda\mathbf{X}\rangle \geq 0$, and by definition,

$$\mathcal{K}^*(\lambda\mathbf{X}) + \mathcal{K}(\mathbf{0}) = \sup_{\mathbf{Y}\in\mathbb{X}} (\langle\lambda\mathbf{X}, \mathbf{Y}\rangle - \mathcal{K}(\mathbf{Y})) + \mathcal{K}(\mathbf{0}) \geq \langle\lambda\mathbf{X}, \mathbf{0}\rangle - \mathcal{K}(\mathbf{0}) + \mathcal{K}(\mathbf{0}) = 0.$$

Combining these inequalities shows that $\mathcal{H}(\mathbf{X}, \mathbf{M}) \geq 0$.

Let $\widetilde{\mathbf{M}} := \mathbf{M} - \epsilon(\gamma\mathbf{M} + \alpha\nabla\mathcal{F}(\mathbf{X}))$. By Lemma 1, we have

$$0 \geq \left\langle -\widetilde{\mathbf{M}} + \nabla\mathcal{K}^*(\lambda\mathbf{X}), \nabla\mathcal{K}(\widetilde{\mathbf{M}}) - \lambda\mathbf{X} \right\rangle$$
$$= \left\langle \epsilon\alpha\nabla\mathcal{F}(\mathbf{X}) - (1 - \epsilon\gamma)\mathbf{M} + \nabla\mathcal{K}^*(\lambda\mathbf{X}), \nabla\mathcal{K}(\widetilde{\mathbf{M}}) - \lambda\mathbf{X} \right\rangle$$
$$0 \geq \left\langle \mathbf{M} - \widetilde{\mathbf{M}}, \nabla\mathcal{K}(\widetilde{\mathbf{M}}) - \nabla\mathcal{K}(\mathbf{M}) \right\rangle$$
$$= \left\langle \epsilon\alpha\nabla\mathcal{F}(\mathbf{X}) + \epsilon\gamma\mathbf{M}, (\nabla\mathcal{K}(\widetilde{\mathbf{M}}) - \lambda\mathbf{X}) - (\nabla\mathcal{K}(\mathbf{M}) - \lambda\mathbf{X}) \right\rangle. \tag{28}$$

By straightforward computation,

$$
\begin{aligned}
\frac{\mathrm{d}}{\mathrm{d}t}\mathcal{H}(\mathbf{X},\mathbf{M}) &= \left\langle \nabla_{\mathbf{X}}\mathcal{H}(\mathbf{X},\mathbf{M}),\dot{\mathbf{X}}\right\rangle + \left\langle \nabla_{\mathbf{M}}\mathcal{H}(\mathbf{X},\mathbf{M}),\dot{\mathbf{M}}\right\rangle \\
&= \left\langle \alpha\nabla\mathcal{F}(\mathbf{X}) + \gamma\nabla\mathcal{K}^*(\lambda\mathbf{X}) + \frac{1-\epsilon\gamma}{1+\epsilon\lambda}(\lambda\nabla\mathcal{K}^*(\lambda\mathbf{X}) - \lambda\mathbf{M}),\nabla\mathcal{K}(\widetilde{\mathbf{M}}) - \lambda\mathbf{X}\right\rangle \\
&\quad + \frac{1-\epsilon\gamma}{1+\epsilon\lambda}\left\langle \nabla\mathcal{K}(\mathbf{M}) - \lambda\mathbf{X}, -\alpha\nabla\mathcal{F}(\mathbf{X}) - \gamma\mathbf{M}\right\rangle \\
&= \frac{\lambda+\gamma}{1+\epsilon\lambda}\left\langle \epsilon\alpha\nabla\mathcal{F}(\mathbf{X}) - (1-\epsilon\gamma)\mathbf{M} + \nabla\mathcal{K}^*(\lambda\mathbf{X}),\nabla\mathcal{K}(\widetilde{\mathbf{M}}) - \lambda\mathbf{X}\right\rangle \\
&\quad + \frac{1-\epsilon\gamma}{\epsilon(1+\epsilon\lambda)}\left\langle \epsilon\alpha\nabla\mathcal{F}(\mathbf{X}) + \epsilon\gamma\mathbf{M}, (\nabla\mathcal{K}(\widetilde{\mathbf{M}}) - \lambda\mathbf{X}) - (\nabla\mathcal{K}(\mathbf{M}) - \lambda\mathbf{X})\right\rangle \\
&\leq 0,
\end{aligned}
$$

where the last line uses (28). $\qquad\square$

We recover (9) by setting $\alpha = \gamma = 1$ in (26). Although an important result, Theorem 7 cannot be directly applied to MUON due to the nondifferentiability of the nuclear norm.

## B  Deferred proofs

**Proposition 2.** $\mathbf{X}^\star \in \mathbb{X}$ *is a KKT point of* (5) *if and only if* $\|\lambda\mathbf{X}^\star\|_{\mathrm{op}} \leq 1$ *and* $\mathcal{S}(\mathbf{X}^\star) = 0$.

*Proof.* Suppose the KKT conditions for the original problem (5) are satisfied, i.e. there exist $\mu \in \mathbb{R}_{\geq 0}$ and a subgradient $\mathbf{G}$ of $\|\cdot\|_{\mathrm{op}}$ at $\mathbf{X}^\star$, where $\|\lambda\mathbf{X}^\star\|_{\mathrm{op}} \leq 1$, such that

$$
\nabla\mathcal{F}(\mathbf{X}^\star) + \mu\mathbf{G} = \mathbf{0} \quad \text{and} \quad \mu(\|\lambda\mathbf{X}^\star\|_{\mathrm{op}} - 1) = 0.
$$

$\|\lambda\mathbf{X}^\star\|_{\mathrm{op}} \leq 1$ is satisfied by primal feasibility. If $\mu = 0$, then $\nabla\mathcal{F}(\mathbf{X}^\star) = \mathbf{0}$, which implies $\mathcal{S}(\mathbf{X}^\star) = 0$. Otherwise, we have $\|\lambda\mathbf{X}^\star\|_{\mathrm{op}} = 1$ by complementary slackness. Let the multiplicity of $\boldsymbol{\sigma}_1(\mathbf{X}^\star)$ be $t$, with corresponding singular vectors $\mathbf{U}_1$ and $\mathbf{V}_1$. Using Watson (1992)'s characterization of the subgradients of $\|\cdot\|_{\mathrm{op}}$, we have $\mathbf{G} = \mathbf{U}_1\mathbf{H}\mathbf{V}_1^\top$, where $\mathbf{H} \in \mathbb{S}_{\succeq\mathbf{0}}^{t\times t}$ and $\mathrm{Tr}(\mathbf{H}) = 1$. Thus $\|\mathbf{G}\|_{\mathrm{tr}} = 1$, and

$$
\mathcal{S}(\mathbf{X}^\star) = \|\nabla\mathcal{F}(\mathbf{X}^\star)\|_{\mathrm{tr}} + \langle\lambda\mathbf{X}^\star,\nabla\mathcal{F}(\mathbf{X}^\star)\rangle = \mu\|\mathbf{G}\|_{\mathrm{tr}} - \mu\langle\lambda\mathbf{X}^\star,\mathbf{G}\rangle = \mu\|\mathbf{G}\|_{\mathrm{tr}} - \mu\|\lambda\mathbf{X}^\star\|_{\mathrm{op}} = 0,
$$

where the third equality uses Lemma 2. $\qquad\square$

**Lemma 1.** *Let* $\mathcal{K},\mathcal{K}^* : \mathbb{X} \to \mathbb{R} \cup \{\infty\}$ *be a convex, closed, and proper pair of conjugate functions with subgradients* $\nabla\mathcal{K}$ *and* $\nabla\mathcal{K}^*$. *Then for all* $\mathbf{X},\mathbf{Y} \in \mathbb{X}$,

$$
\langle\nabla\mathcal{K}(\mathbf{X}) - \nabla\mathcal{K}(\mathbf{Y}),\mathbf{X} - \mathbf{Y}\rangle \geq 0 \tag{19}
$$

$$
\langle\nabla\mathcal{K}(\mathbf{X}) - \mathbf{Y},\mathbf{X} - \nabla\mathcal{K}^*(\mathbf{Y})\rangle \geq 0. \tag{20}
$$

*Proof.* By definition of subgradients, we have

$$
\begin{aligned}
\mathcal{K}(\mathbf{Y}) - \mathcal{K}(\mathbf{X}) &\geq \langle\nabla\mathcal{K}(\mathbf{X}),\mathbf{Y} - \mathbf{X}\rangle \\
\mathcal{K}(\mathbf{X}) - \mathcal{K}(\mathbf{Y}) &\geq \langle\nabla\mathcal{K}(\mathbf{Y}),\mathbf{X} - \mathbf{Y}\rangle.
\end{aligned}
$$

Summing the inequalities gives $0 \geq \langle\nabla\mathcal{K}(\mathbf{Y}) - \nabla\mathcal{K}(\mathbf{X}),\mathbf{X} - \mathbf{Y}\rangle$, which shows (19). (20) follows by setting $\mathbf{Y} \leftarrow \nabla\mathcal{K}^*(\mathbf{Y})$ in (19) and using $\mathbf{Y} \in \partial\mathcal{K}(\nabla\mathcal{K}^*(\mathbf{Y}))$. $\qquad\square$

**Lemma 2.** *Let* $\mathcal{K}(\mathbf{X}) = \|\mathbf{X}\|$ *for a norm* $\|\cdot\|$ *on* $\mathbb{X}$. *Then* $\langle\nabla\mathcal{K}(\mathbf{X}),\mathbf{X}\rangle = \mathcal{K}(\mathbf{X})$.

*Proof.* By definition of a subgradient and properties of norms,

$$0 = \mathcal{K}(\mathbf{0}) \geq \mathcal{K}(\mathbf{X}) + \langle \nabla\mathcal{K}(\mathbf{X}), \mathbf{0} - \mathbf{X} \rangle = \mathcal{K}(\mathbf{X}) - \langle \nabla\mathcal{K}(\mathbf{X}), \mathbf{X} \rangle$$
$$2\mathcal{K}(\mathbf{X}) = \mathcal{K}(2\mathbf{X}) \geq \mathcal{K}(\mathbf{X}) + \langle \nabla\mathcal{K}(\mathbf{X}), 2\mathbf{X} - \mathbf{X} \rangle = \mathcal{K}(\mathbf{X}) + \langle \nabla\mathcal{K}(\mathbf{X}), \mathbf{X} \rangle \,.$$

Combining the inequalities shows that $\langle \nabla\mathcal{K}(\mathbf{X}), \mathbf{X} \rangle = \mathcal{K}(\mathbf{X})$. $\qquad\square$

**Lemma 3.** *In the setting of Proposition 3, let $\mathcal{K}(\mathbf{X}) = \|\mathbf{X}\|_{\mathrm{tr}}$, $\nabla\mathcal{K}(\mathbf{X}) = \mathrm{msgn}(\mathbf{X})$, $\|\lambda\mathbf{X}_0\|_{\mathrm{op}} \leq 1$, $\eta_t = \eta$, and $C_\mathcal{K} := \sqrt{\min(n,m)}$. Then for all $t > 0$,*

$$\left\| \nabla\mathcal{F}(\mathbf{X}_t) + \widetilde{\mathbf{M}}_t \right\|_{\mathrm{F}} \leq \frac{2\eta C_\mathcal{K} L(1 + \beta_1 - \beta_2)}{1 - \beta_2} + \beta_1 \beta_2^{t-1} \|\nabla\mathcal{F}(\mathbf{X}_0) + \mathbf{M}_0\|_{\mathrm{F}} \,.$$

*Proof.* By Proposition 1, we have $\|\lambda\mathbf{X}_t\|_{\mathrm{op}} \leq 1$ for all $t \geq 0$. Furthermore, for all $t > 0$,

$$\begin{aligned}
\|\nabla\mathcal{F}(\mathbf{X}_t) - \nabla\mathcal{F}(\mathbf{X}_{t-1})\|_{\mathrm{F}} &\leq L \|\mathbf{X}_t - \mathbf{X}_{t-1}\|_{\mathrm{F}} = \eta_{t-1} L \left\| \mathrm{msgn}(\widetilde{\mathbf{M}}_t) - \lambda\mathbf{X}_t \right\|_{\mathrm{F}} \\
&\leq \eta_{t-1} C_\mathcal{K} L \left\| \mathrm{msgn}(\widetilde{\mathbf{M}}_t) - \lambda\mathbf{X}_t \right\|_{\mathrm{op}} \leq 2\eta_{t-1} C_\mathcal{K} L,
\end{aligned} \tag{29}$$

where the first line uses smoothness. Recalling (3),

$$\begin{aligned}
\|\nabla\mathcal{F}(\mathbf{X}_t) + \mathbf{M}_t\|_{\mathrm{F}} &= \|\nabla\mathcal{F}(\mathbf{X}_t) + \beta_2\mathbf{M}_{t-1} - (1-\beta_2)\nabla\mathcal{F}(\mathbf{X}_{t-1})\|_{\mathrm{F}} \\
&= \|\nabla\mathcal{F}(\mathbf{X}_t) - \nabla\mathcal{F}(\mathbf{X}_{t-1}) + \beta_2(\nabla\mathcal{F}(\mathbf{X}_{t-1}) + \mathbf{M}_{t-1})\|_{\mathrm{F}} \\
&= \left\| \sum_{k=1}^t \beta_2^{t-k}(\nabla\mathcal{F}(\mathbf{X}_k) - \nabla\mathcal{F}(\mathbf{X}_{k-1})) + \beta_2^t(\nabla\mathcal{F}(\mathbf{X}_0) + \mathbf{M}_0) \right\|_{\mathrm{F}} \\
&\leq \sum_{k=1}^t \beta_2^{t-k} \|\nabla\mathcal{F}(\mathbf{X}_k) - \nabla\mathcal{F}(\mathbf{X}_{k-1})\|_{\mathrm{F}} + \beta_2^t \|\nabla\mathcal{F}(\mathbf{X}_0) + \mathbf{M}_0\|_{\mathrm{F}} \\
&\leq 2C_\mathcal{K} L \sum_{k=1}^t \beta_2^{t-k} \eta_{k-1} + \beta_2^t \|\nabla\mathcal{F}(\mathbf{X}_0) + \mathbf{M}_0\|_{\mathrm{F}} \,,
\end{aligned} \tag{30}$$

where the third line iterates and expands the first two lines, the fourth line uses the triangle inequality, and the fifth line uses (29). Thus

$$\begin{aligned}
\left\| \nabla\mathcal{F}(\mathbf{X}_t) + \widetilde{\mathbf{M}}_t \right\|_{\mathrm{F}} &= \|\nabla\mathcal{F}(\mathbf{X}_t) + \beta_1\mathbf{M}_{t-1} - (1-\beta_1)\nabla\mathcal{F}(\mathbf{X}_{t-1})\|_{\mathrm{F}} \\
&= \|\nabla\mathcal{F}(\mathbf{X}_t) - \nabla\mathcal{F}(\mathbf{X}_{t-1}) + \beta_1(\nabla\mathcal{F}(\mathbf{X}_{t-1}) + \mathbf{M}_{t-1})\|_{\mathrm{F}} \\
&\leq \|\nabla\mathcal{F}(\mathbf{X}_t) - \nabla\mathcal{F}(\mathbf{X}_{t-1})\|_{\mathrm{F}} + \beta_1 \|\nabla\mathcal{F}(\mathbf{X}_{t-1}) + \mathbf{M}_{t-1}\|_{\mathrm{F}} \\
&\leq 2\eta_{t-1} C_\mathcal{K} L + \beta_1 \left( 2C_\mathcal{K} L \sum_{k=1}^{t-1} \beta_2^{t-k-1} \eta_{k-1} + \beta_2^{t-1} \|\nabla\mathcal{F}(\mathbf{X}_0) + \mathbf{M}_0\|_{\mathrm{F}} \right) \\
&= 2\eta_{t-1} C_\mathcal{K} L + 2\beta_1 C_\mathcal{K} L \sum_{k=1}^{t-1} \beta_2^{t-k-1} \eta_{k-1} + \beta_1 \beta_2^{t-1} \|\nabla\mathcal{F}(\mathbf{X}_0) + \mathbf{M}_0\|_{\mathrm{F}} \,,
\end{aligned} \tag{31}$$

where the third line uses the triangle inequality and the fourth line uses (29) and (30). The result follows upon setting $\eta_t = \eta$ and using $\sum_{k=1}^{t-1} \beta_2^{t-k-1} \leq \sum_{j=0}^\infty \beta_2^j = \frac{1}{1-\beta_2}$. $\qquad\square$

**Lemma 4.** *Let $\mathcal{K}(\mathbf{X}) = \|\mathbf{X}\|$ for a norm $\|\cdot\|$ on $\mathbb{X}$. Then for all $\mathbf{X} \in \mathbb{X}$, $\mathcal{K}^*(\nabla\mathcal{K}(\mathbf{X})) = 0$.*

*Proof.* By definition of a subgradient and Lemma 2,

$$\mathcal{K}(\mathbf{Y}) \geq \mathcal{K}(\mathbf{X}) + \langle \nabla\mathcal{K}(\mathbf{X}), \mathbf{Y} - \mathbf{X} \rangle = \langle \nabla\mathcal{K}(\mathbf{X}), \mathbf{Y} \rangle$$

for all $\mathbf{Y} \in \mathbb{X}$, which implies that

$$0 \leq \|\nabla\mathcal{K}(\mathbf{X})\|_* = \sup_{\mathbf{Y} \neq \mathbf{0}} \frac{\langle \nabla\mathcal{K}(\mathbf{X}), \mathbf{Y} \rangle}{\mathcal{K}(\mathbf{Y})} \leq 1$$

by definition of the dual norm. We conclude that $\mathcal{K}^*(\nabla\mathcal{K}(\mathbf{X})) = 0$ by Fact 1. $\square$

**Lemma 5.** *Let $\mathbf{X}, \mathbf{Y}$ be $\mathbb{X}$-valued random variables satisfying $\mathrm{Var}(\mathbf{Y}) < \infty$, and let $\mathcal{K}(\mathbf{X}) = \|\mathbf{X}\|$ for a norm $\|\cdot\|$ on $\mathbb{X}$. Then there exists a constant $C_\mathcal{K}$ such that*

$$\mathbb{E}\left[\langle \mathbb{E}[\mathbf{Y}] - \mathbf{Y}, \nabla\mathcal{K}(\mathbf{X} + \epsilon\mathbf{Y}) \rangle\right] \leq C_\mathcal{K}\sqrt{\mathrm{Var}(\mathbf{Y})}.$$

*Proof.* By the Fenchel–Young inequality and Lemma 4,

$$\langle \mathbb{E}[\mathbf{Y}] - \mathbf{Y}, \nabla\mathcal{K}(\mathbf{X} + \epsilon\mathbf{Y}) \rangle \leq \mathcal{K}(\mathbb{E}[\mathbf{Y}] - \mathbf{Y}) + \mathcal{K}^*(\nabla\mathcal{K}(\mathbf{X} + \epsilon\mathbf{Y})) = \mathcal{K}(\mathbb{E}[\mathbf{Y}] - \mathbf{Y}).$$

By the equivalence of norms on finite-dimensional vector spaces, there exists a constant $C_\mathcal{K}$ such that $\mathcal{K}(\mathbf{X}) \leq C_\mathcal{K}\|\mathbf{X}\|_\mathrm{F}$. Taking expectations,

$$\mathbb{E}\left[\langle \mathbb{E}[\mathbf{Y}] - \mathbf{Y}, \nabla\mathcal{K}(\mathbf{X} + \epsilon\mathbf{Y}) \rangle\right] \leq \mathbb{E}\left[\mathcal{K}(\mathbb{E}[\mathbf{Y}] - \mathbf{Y})\right] \leq C_\mathcal{K}\mathbb{E}\left[\|\mathbb{E}[\mathbf{Y}] - \mathbf{Y}\|_\mathrm{F}\right]$$

$$\leq C_\mathcal{K}\sqrt{\mathbb{E}\left[\|\mathbb{E}[\mathbf{Y}] - \mathbf{Y}\|_\mathrm{F}^2\right]} = C_\mathcal{K}\sqrt{\mathrm{Var}(\mathbf{Y})},$$

where the second line uses Jensen's inequality. $\square$

**Lemma 6.** *Let $\mathbf{X}, \mathbf{Y}$ be $\mathbb{X}$-valued random variables satisfying*

$$\mathrm{Var}(\mathbf{X}) \leq \sigma^2, \ \mathrm{Var}(\mathbf{Y}) \leq \sigma^2, \ and \ \|\mathbb{E}[\mathbf{X}] - \mathbb{E}[\mathbf{Y}]\|_\mathrm{F} \leq R.$$

*Then $\mathbb{E}[\|\mathbf{X} - \mathbf{Y}\|_\mathrm{F}] \leq 2\sigma + R$.*

*Proof.* We have

$$\mathbb{E}[\|\mathbf{X} - \mathbf{Y}\|_\mathrm{F}] \leq \mathbb{E}[\|\mathbf{X} - \mathbb{E}[\mathbf{X}]\|_\mathrm{F} + \|\mathbb{E}[\mathbf{X}] - \mathbb{E}[\mathbf{Y}]\|_\mathrm{F} + \|\mathbf{Y} - \mathbb{E}[\mathbf{Y}]\|_\mathrm{F}]$$

$$\leq \sqrt{\mathbb{E}\left[\|\mathbf{X} - \mathbb{E}[\mathbf{X}]\|_\mathrm{F}^2\right]} + R + \sqrt{\mathbb{E}\left[\|\mathbf{Y} - \mathbb{E}[\mathbf{Y}]\|_\mathrm{F}^2\right]} \leq 2\sigma + R,$$

where the first line uses the triangle inequality and the second line uses Jensen's inequality. $\square$

**Lemma 7.** *Let $\mathbf{G}$, $\mathbf{X}$, and $\mathbf{Y}$ be $\mathbb{X}$-valued random variables satisfying $\mathbb{E}[\mathbf{G} \mid \mathbf{X}] = \mathbf{Y}$. Then*

$$\mathbb{E}\left[\|\mathbf{Y}\|_\mathrm{tr} + \langle \lambda\mathbf{X}, \mathbf{Y} \rangle\right] \leq \mathbb{E}[\|\mathbf{G}\|_\mathrm{tr} + \langle \lambda\mathbf{X}, \mathbf{G} \rangle].$$

*Proof.* Trivial by Jensen's. $\square$

**Lemma 8.** *Let $X$ be a nonnegative random variable such that $\mathbb{E}[X] = 0$. Then $X = 0$ a.s.*

*Proof.* By the layer cake representation,

$$0 = \mathbb{E}[X] = \int_0^\infty \Pr(X > t)\mathrm{d}t,$$

so $\Pr(X > t) = 0$ for (Lebesgue) almost every $t > 0$. Since $\Pr(X > t)$ is a right-continuous function of $t$, we have that $\Pr(X > t) = 0$ for all $t > 0$. The conclusion follows from

$$\Pr(X > 0) = \lim_{t \searrow 0} \Pr(X > t) = 0.$$

$\square$

**Lemma 9** (LaSalle's invariance principle for stochastic dynamical systems)**.** *Let $\{\mathbf{X}_t\}_{t\in\mathbb{N}}$ be a stochastic process contained in a bounded set a.s., and suppose there exist a nonnegative function $\mathcal{V}$ and a nonnegative, lower semicontinuous function $h$ such that*

$$\mathbb{E}[\mathcal{V}(\mathbf{X}_{t+1}) \mid \mathcal{F}_t] - \mathcal{V}(\mathbf{X}_t) \leq -\alpha_t h(\mathbf{X}_{t+\ell}) + \gamma_t \ \ a.s.,$$

*where $\{\mathcal{F}_t\}_{t\in\mathbb{N}}$ is the natural filtration of $\{\mathbf{X}_t\}_{t\in\mathbb{N}}$, $\ell \in \mathbb{N}$, and the nonnegative sequences $\{\alpha_t\}_{t\in\mathbb{N}}$ and $\{\gamma_t\}_{t\in\mathbb{N}}$ satisfy*

$$\sum_{t=0}^{\infty} \alpha_t = \infty \quad and \quad \sum_{t=0}^{\infty} \gamma_t < \infty.$$

*Let $\mathbb{M}$ be the $\omega$-limit set of $\{\mathbf{X}_t\}_{t\in\mathbb{N}}$. Then $\mathbb{M}$ is contained in the set $\{\mathbf{X} \in \mathbb{X} \mid h(\mathbf{X}) = 0\}$.*

*Proof.* Define the auxiliary function

$$\widehat{\mathcal{V}}_t := \mathcal{V}(\mathbf{X}_t) - g_t, \quad \text{where } g_t := \sum_{s=0}^{t-1} \gamma_s.$$

Then, we have

$$\mathbb{E}\left[\widehat{\mathcal{V}}_{t+1} \mid \mathcal{F}_t\right] - \widehat{\mathcal{V}}_t = \mathbb{E}[\mathcal{V}(\mathbf{X}_{t+1}) \mid \mathcal{F}_t] - \mathcal{V}(\mathbf{X}_t) - \gamma_t \leq -\alpha_t h(\mathbf{X}_{t+\ell}) \leq 0 \text{ a.s.} \tag{32}$$

By (32), the nonnegativity of $\mathcal{V}$, and $\lim_{t\to\infty} g_t < \infty$, it follows that $\{\widehat{\mathcal{V}}_t\}_{t\in\mathbb{N}}$ is a supermartingale with $\sup_{t\in\mathbb{N}} \mathbb{E}\left[\widehat{\mathcal{V}}(\mathbf{X}_t)^-\right] < \infty$. By Doob's supermartingale convergence theorem, we conclude that $\{\widehat{\mathcal{V}}_t\}_{t\in\mathbb{N}}$ and hence $\{\mathcal{V}(\mathbf{X}_t)\}_{t\in\mathbb{N}}$ converges almost surely to a random variable with finite expectation.

Now, taking expectations of (32) and summing from $t = 0$ to $\infty$, we obtain

$$\sum_{t=0}^{\infty} \mathbb{E}[\alpha_t h(\mathbf{X}_{t+\ell})] = \sum_{t=\ell}^{\infty} \alpha_{t-\ell} \mathbb{E}[h(\mathbf{X}_t)] < \infty.$$

Since $\sum_{t=0}^{\infty} \alpha_t = \infty$, this implies $\liminf_{t\to\infty} \mathbb{E}[h(\mathbf{X}_t)] = 0$. By Fatou's lemma, we have

$$0 \leq \mathbb{E}\left[\liminf_{t\to\infty} h(\mathbf{X}_t)\right] \leq \liminf_{t\to\infty} \mathbb{E}[h(\mathbf{X}_t)] = 0$$

and hence

$$\liminf_{t\to\infty} h(\mathbf{X}_t) = 0 \text{ a.s.}$$

by Lemma 8. By the lower semicontinuity and nonnegativity of $h$ and the (almost sure) boundedness of $\mathbf{X}_t$, the $\omega$-limit set $\mathbb{M}$ of $\{\mathbf{X}_t\}_{t\in\mathbb{N}}$ must satisfy $h(\mathbf{X}) = 0$ for all $\mathbf{X} \in \mathbb{M}$, i.e.

$$\mathbb{M} \subseteq \{\mathbf{X} \in \mathbb{X} \mid h(\mathbf{X}) = 0\}.$$

$\square$

