# OpenReview forum: "Muon Optimizes Under Spectral Norm Constraints"
_TMLR — Accepted by TMLR_

### Review · Reviewer_vroB · 2026-01-05

**Summary Of Contributions:**

This paper provides a theoretical convergence analysis of the Muon optimizer, which has recently shown promising empirical performance. The authors approach this by first framing Muon as a special case of the Lion-K family of optimizers, with K corresponding to the trace norm of the matrix parameter. This framing allows them to leverage the theoretical framework developed in (Chen et al. 2024) to analyze the convergence of Muon and to show that Muon optimizes the objective function under implicit constraints on the singular values of the parameter.

To support their theory, the paper presents empirical results on toy matrix optimization problems as well as image classification with different architectures, to verify that the solutions found by Muon satisfy the implicit singular-value constraints. They also further illustrate the difference in the singular value distribution when using Muon compared to AdamW.

While the reduction of Muon to a Lion-K instance is clean and well presented, my main concern is that the theoretical results appear to be largely a special case of the existing results in Chen et al. (2024), rather than an extension as claimed. I expand on this concern below.

**Audience:**

Yes

**Audience Explanation:**

The Muon optimizer has received recent attention, and a principled theoretical understanding of its behavior or insights into its performance is definitely of interest to the TMLR audience

**Claims And Evidence:**

No

**Claims Explanation:**

The paper’s central claim is that it extends the convergence analysis of Lion-K optimizers (in Chen et al (2024)) to Muon, where the challenge is extending the results to handle the non-differentiable trace norm. However, Section 4 of Chen et al. (2024) already provides a discrete-time convergence analysis for Lion-K optimizers that applies to non-differentiable K (theorem 4.1 for the deterministic case and the stochastic analysis in the appendix), which shows the convergence rate to the domain of the conjugate function of K* and also the rate of convergence to the stationary points of the corresponding implicit constrained optimization. Based on my reading, even the proof steps in the main body seem to closely follow the same steps in the earlier work. As a result, currently it is unclear to me in what precise sense the analysis constitutes an extension, rather than a direct application of existing theory to a particular choice of K.

It is possible that I am missing a subtle point where the analysis in Chen et al. (2024) does not apply to Muon, or where the present paper provides additional insight that is not already implied by plugging in the specific K in the earlier results. I find it hard to parse this distinction in the current presentation. If the theory indeed requires additional arguments or addresses a gap in the prior analysis, it is helpful to have this explicitly discussed and clarified. Otherwise, the claims and framing of the results should be adjusted to reflect the insights provided by the paper for the Muon optimizer.

**Requested Changes:**

Given that the main focus of the paper is on the theoretical analysis, I suggest reframing the organization of the paper to better reflect and discuss which parts of the analysis in Chen et al (2024) are specifically extended/changed to handle the case of Muon (Please see the discussion on the claims).

---

> ### Author Response · Authors · 2026-01-23
>
> We thank the reviewer for their thorough evaluation and constructive feedback. We will state the technical contributions of this work more precisely in a revision. In particular, we build upon the analysis of [CLLL24] in the following ways:
> - We generalize the Lion-$\mathcal{K}$ family to handle matrix-valued parameters. This is a fairly straightforward extension of the vector-valued case, but it is nevertheless important to show that it can be done.
> - The stochastic analysis result (Theorem B.14) in [CLLL24] relies on the assumption that $\mathcal{K}$ has a Lipschitz weak gradient, which is not the case when $\mathcal{K}$ is taken to be nuclear norm. In contrast, our stochastic analysis result (Theorem 4) removes this restrictive assumption, ensuring that the results hold for the nuclear norm.
> - The reviewer claims that the results of [CLLL24] show "convergence to the stationary points of the corresponding implicit constrained optimization", but the convergence rates are given in terms of nonstandard stationarity measures denoted by $\Delta_t^1$ and $\Delta_t^2$ in that work. Additional analysis is required to establish a notion of convergence to KKT points, which is provided by Proposition 2 in our work.
> - Theorem 5 and Theorem 6 in this work provide convergence results in the nonconstant, Robbins-Monro-style step size regime, which was not explored in [CLLL24].
>
> It is true that the proofs in this work closely follow the proofs in [CLLL24]. This is unsurprising, considering that the key insight of this work is that Muon belongs to the Lion-$\mathcal{K}$ family of optimizers.

---

> > ### Comment · Reviewer_vroB · 2026-02-09
> >
> > Thank you for the response and the helpful clarifications. I suggest highlighting these points in the revision to better clarify the distinctions between your theory and existing frameworks.

---

### Review · Reviewer_bGMr · 2026-01-07

**Summary Of Contributions:**

This paper presents a theoretical interpretation of the Muon optimizer by showing that it is a special case of the Lion-K framework with the nuclear norm as the underlying convex potential. Under this view, Muon with decoupled weight decay implicitly solves a spectral-norm-constrained optimization problem. The authors establish convergence rates and almost sure convergence to KKT points using Lyapunov analysis, and show that the framework naturally generalizes Muon to a broader class of spectral optimizers.

Strengths
- The paper is well-structured and technically sound, with clear connections to prior optimization theory.


- The paper shows that Muon with decoupled weight decay does not merely penalize large singular values but instead enforces a hard spectral norm constraint, as evidenced by the conjugate-form objective, infinite cost outside the feasible set, exponential contraction into the constraint ball, and convergence to KKT points rather than unconstrained stationary points.


Weaknesses
- The overall approach is largely derivative of the Lion-K framework, and the conceptual novelty beyond specializing and extending existing theory may be limited.

- Some technical sections are theory-dense, which may reduce accessibility for readers not already familiar with Lion-K or Lyapunov-based analyses.

**Audience:**

Yes

**Audience Explanation:**

The paper will be of interest to researchers working on optimization theory and implicit regularization in deep learning, particularly those studying adaptive and momentum-based methods. Its theoretical interpretation of the Muon optimizer and its connection to constrained optimization provides insights relevant to the TMLR audience.

**Broader Impact Concerns:**

This work is mainly theoretical and does not concern major ethical implications.

**Claims And Evidence:**

Yes

**Claims Explanation:**

The main claims are supported by formal convergence proofs within the Lion-K framework, including explicit convergence rates and almost sure convergence in both deterministic and stochastic settings. The assumptions are standard and clearly stated, and the theoretical results are consistent with the provided empirical illustrations.

**Requested Changes:**

- Please remove the proofs from the main paper to make the readability of theoretical results better.

- Please discuss the connection between your work and the methods/results presented in https://arxiv.org/pdf/2503.12645.

- Can you clarify the novelty of your stochastic convergence results?
In particular, is there an analogous result in the Lion-K paper similar to the deterministic case, or does your work establish genuinely new guarantees in the stochastic setting?

---

> ### Author Response · Authors · 2026-01-23
>
> We thank the reviewer for their thorough evaluation and constructive feedback. We will move the more technical parts to the appendix in a revision.
>
> Apart from broadly providing theoretical guarantees for Muon, we believe that our results are largely incomparable with [Kov25]. In particular, we consider a generalized version of Muon that incorporates Nesterov momentum as a special case of Lion-$\mathcal{K}$, while [Kov25] analyzes the algorithm from the perspective of a non-Euclidean trust-region method. Furthermore, the convergence rates in [Kov25] and our work are given for different stationarity measures, and the results of [Kov25] are dependent on restrictive settings for the momentum and weight decay hyperparameters.
>
> We refer the reviewer to the discussion with Reviewer vroB for clarification on the technical contributions.

---

> > ### Comment · Reviewer_bGMr · 2026-02-10
> >
> > Thanks for your response and clarification.

---

### Review · Reviewer_6Xmk · 2026-01-11

**Summary Of Contributions:**

This paper investigates the connection between two optimizers, Muon and Lion-$\mathcal{K}$. In particular, Lion-$\mathcal{K}$ uses a convex function $\mathcal{K}$ to perform preconditioning by mapping the momentum $\widetilde{M}_t$ to $\nabla \mathcal{K}(\widetilde{M}_t)$. When $\mathcal{K}$ is chosen to be the spectral norm, this preconditioning mapping $\nabla \mathcal{K}$ reduces exactly to matrix sign. Therefore, Muon can be viewed as Lion-$\mathcal{K}$ with function $\mathcal{K}$ being the spectral norm.

In addition, this paper shows that a variant of Muon with decoupled weight decaying can find an KKT solution of the spectral norm constrained optimization problem.

**Audience:**

Yes

**Audience Explanation:**

This work treats Muon as a special instance of Lion-$\mathcal{K}$ and analyzes it using a theoretical framework for Muon ("Lion Secretly Solves Constrained Optimization"), yielding a practical version of Muon with convergence guarantees. Furthermore, it demonstrates that Muon with weight decay implicitly enforces a spectral-norm constraint, which is of interest to researchers studying Muon.

**Claims And Evidence:**

Yes

**Claims Explanation:**

This paper clearly explains the relationship between Muon and Lion-K. The established convergence results consist of two phases: Phase 1, convergence to feasibility, and Phase 2, convergence to stationarity. These convergence results exhibit a pattern similar to "Lion Secretly Solves Constrained Optimization".

**Requested Changes:**

Major Comments:
1) Page 7. I do not understand why $\widetilde{M}$ is said to be obtained using Nesterov momentum. It seems that $\widetilde{M}$ and $M$ are updated using the same scheme. As I understand it, Polyak momentum (PM) and Nesterov momentum (NM) differ in the point at which the gradient is evaluated: PM takes the gradient at the current iterate $X_t$, while NM takes the gradient at a lookahead point $X_t + \eta M_t$. Therefore, from my perspective, $\widetilde{M}$ is also obtained using Polyak momentum, but with a different momentum parameter.

2) Page 9. The feasibility guarantee is provided in Proposition 1. Later, $\{\eta_t\}$ is chosen to depend on the maximum number of iterations $T$ (see Theorems 3 and 4). Can you show that the algorithm can find a nearly feasible point with this choice of $\eta_t$.

3) Page 10. Global convergence is established as a consequence of Lemma 9 (LaSalle's invariance principle). Is feasibility only guaranteed almost surely in the limit? It seems to me that feasibility converges deterministically at a certain rate, according to Proposition 1.

4) Page 14. It seems the momentum parameters $\beta_1$ and $\beta_2$ are selected to be constant. This differs from the analysis of SGD with momentum for nonconvex optimization; for example, see "Momentum Improves Normalized SGD". Please comment on this.


Minor Comments:
1) Pages 3-4. Please add a reference to Proposition 2 for the proof of the statement that 'satisfying the KKT conditions is equivalent to the score function being zero and the solution being feasible'.

2) Page 17. Please add a reference to the LaSalle's invariance principle.

---

> ### Author Response · Authors · 2026-01-23
>
> We thank the reviewer for their thorough evaluation and constructive feedback. We will incorporate the suggested references in a revision.
>
> Regarding the major comments,
> 1. We refer the reviewer to Appendix B.2 of the Lion-$\mathcal{K}$ paper [CLLL24] for a discussion of how the Lion-$\mathcal{K}$ update rule incorporates Nesterov momentum.
> 2. In the settings of Theorem 3 and Theorem 4, we require $\eta_t=\eta=\Theta\left(\frac{1}{\sqrt{T}}\right)$ to be a constant that depends on $T$. Thus, to ensure $\eta\lambda\leq1$ as well, it suffices to choose, e.g., $\eta=\frac{1}{\lambda\sqrt{T}}$. In regimes of interest, it is usually the case that $\lambda\ll1$ and $T$ is very large, so the $\eta\lambda\leq1$ condition is very mild.
> 3. Proposition 1 shows a linear convergence rate to the feasible set, corresponding to Phase I. Within the feasible set itself, convergence to the set of KKT points, corresponding to Phase II, can only be guaranteed almost surely in the limit. Compare with, e.g., the convergence of the Robbins–Monro algorithm.
> 4. Allowing the momentum coefficients to be dependent on time and/or other hyperparameters is a common approach for obtaining improved convergence rates. We do not require any time-dependent setting of $\beta_1$ or $\beta_2$ for our analysis. We also note that in almost all practical settings, momentum coefficients are set to be constant.

---

### Comment · Action_Editor_cNxs · 2025-11-17

Dear authors,

I'm checking regularly for available reviewers to assign to the paper but there haven't been many available with relevant expertise. Unfortunately, this will delay the review process, most likely by a couple of weeks. I'll keep checking for reviewers and assign them as soon as they are available.

Your AE

---

### Decision · Action_Editor_cNxs · 2026-02-18

**Recommendation:** Accept as is

**Audience:**

Yes

**Audience Explanation:**

The Muon optimizer is currently of interest to many optimization researchers and the paper contributes meaningfully to the existing literature. It would definitely interest a number of optimization experts.

**Claims And Evidence:**

Yes

**Claims Explanation:**

The paper presents a rigorous theory on application of the Lion-K framework with the nuclear norm to Muon. The developed theory is well presented and supported by numerical evaluations on a wide range of problems (several LLMs, ViT, ResNet, and others).